# Remote-controlled mechanical and directional motions of photoswitchable DNA condensates

Hirotake Udono ⑩[1], Shin-ichiro M. Nomura ⑩[2] & Masahiro Takinoue ⑩[1,3] ✉

Membrane-free synthetic DNA-based condensates enable programmable control of dynamic behaviors as shown by phase-separated condensates in biological cells. We demonstrate remote-controlled microflow using photo-controllable state transitions of DNA condensates, assembled from multi-branched DNA nanostructures via sticky-end (SE) hybridization. Introducing azobenzene into SEs enables their photoswitchable binding affinity, which underlies photoreversible fluidity of the resulting condensates that transition between gel/liquid/dissociated states in a wavelength-dependent manner. Leveraging base-sequence programmability, spatially coupled orthogonal DNA condensates with divergent photoresponsive capabilities perform multi-modal mechanical actions that depend on azobenzene insertion sites in the SE, including switching flows radially expanding and converging under photo-switching. Localizing photoswitching within a DNA liquid condensate generates two distinct directional motions, whose contrasting morphology, direction, and lifetime are determined by switching frequency. Numerical simulations reveal its regulatory role in weight-adjusting energy-exchanging and energy-dissipative interactions between the photoirradiated and uni-irradiated domains.

With increasing sophistication of nano-/micro-technology, remotely controllable miniaturized objects have been the subject of interest in various research fields. Most studies have primarily focused on the dynamic adjustment in structure[1,2], motion[3], and function[4] of micrometer-sized objects, such as microgels and liposomes. One disregarded target is the remote controllability of micrometer-sized fluid[5,6]. Notably, there has been an increasing demand for exploitable remote-controlled fluids for various mechanical manipulations within confined environments such as micro-reactors[7], miniature soft robots[1,8], and artificial[9] or biological[5] cells. The potential examples include the deformation of soft microstructures[10] (*e.g.*, gels, droplets, membrane sheets[11], and even micro-wires[12]) and fluid-driven transport using diffusion and convection[5,13]. In the last few decades, microflow regulation has been investigated exclusively using

microchannels that produce the fluid effects to achieve numerous tasks, such as solution mixing[14], droplet generation/manipulation[15], and particle sorting[16]. However, these top-down microfluidic approaches are unsuitable for microflow manipulation within confined environments because of limited accessibility. Consequently, the control mechanism and rational design of remote-controlled microflows remain largely unknown. A rational design principle is to translate molecular-level control into larger-scale microflow behavior, which is realized through the following three mechanisms: (1) An energy-transducing mechanism that receives and converts input energy into mechanical actions. This energy-transducing system should operate under isothermal conditions to limit nonspecific chemical reactions within confined environments. (2) An on/off switching capability that activates or deactivates the remote-

[1]Department of Computer Science, School of Computing, Institute of Science Tokyo, Yokohama, Kanagawa 226-8501, Japan. [2]Department of Robotics, Graduate School of Engineering, Tohoku University, Sendai, Miyagi 980-8579, Japan. [3]Research Center for Autonomous Systems Materialogy (ASMat), Institute of Innovative Research (IIR), Institute of Science Tokyo, Yokohama, Kanagawa 226-8501, Japan. ✉e-mail: takinoue@comp.isct.ac.jp

controlled microflow upon a signal input. (3) Dynamics programmability based on molecularly encoded information.

For the programmable microfluidic systems, exploiting DNA as a design material is an appealing strategy because of the high addressability and programmability of DNA binding sites[17,18]. In the last few decades, the powerful molecular recognition capability of DNA has been exploited to control the self-assembly of highly structured nanosystems[17,19–21]. Beyond nanofeatures, the specific bonding mechanism of DNA has been used to form DNA hydrogels, which are compelling targets for numerous biomedical applications due to their high biocompatibility, easy synthesis, and functionalization[22,23]. Recently, DNA liquid condensates–micro-scale liquid condensates of DNA nanostructures with well-engineered sequences that emerge through phase separation[24–41]–have drawn significant attention as model systems[36] of biological phase-separated liquid condensates[42], which have been extensively explored for their functions[43], organization[44,45], and links to diseases[46]. The molecular-level base-sequence design allows DNA liquid condensates to realize the representative macroscopic behaviors of biological condensates programmably by achieving sequence-specifically directed interaction[24] and high programmability of structures[24,25,29,31], phase behaviors[24,31,47], and physical properties[47–49]. Specifically, DNA liquid condensates exhibit a thermoreversible phase behavior between gel, liquid, and dissociated states, which is prescribed by sequence design[24,50]. However, DNA liquid condensates still lack the remote controllability of isothermal state transition and energy-transducing mechanisms, eagerly anticipated features for intracellular microfluidic control. To satisfy these requirements, photochemical reactions have distinct advantages as a means of effective signal transmission, energy injection, and isothermal control, including minimal invasiveness and high-locality control[51–55]. Previous studies have reported the photoregulated state transitions of DNA self-assembled matter with low spatial locality (owing to high concentration threshold)[56,57] or in low specificity (because of photoswitchable surfactant usage)[58]. Moreover, photoswitchable materials reported thus far, using DNA or other biomolecules[59,60], have achieved no significant mechanical actions, with a primary focus on controlled compaction[61,62] and phase change[34,58,63,64].

In this study, we demonstrate remote-controlled microflow using photoresponsive DNA condensates. They were fabricated through the self-assembly of branched DNA motifs, Y-shaped nanostructures (Y-motifs) of three single-stranded DNAs (ssDNAs), base-paired in the stem region with a sticky end (SE) at each terminal. SE hybridization enables the self-assembly of Y-motifs into micro-scale network-structured DNA condensates. The motifs were

equipped with a photoresponsive capability by introducing azobenzene (Azo), a well-studied photoisomerizable compound[53], into the SEs. Alternating ultraviolet light (UV) and visible light (Vis) irradiation allowed their dissociation and reassociation, leading to a photoreversible fluidity change in the condensates between gel/liquid/dissociated states. To design an energy-transducing system, we cross-linked a photoresponsive DNA motif with a non-photoresponsive DNA motif orthogonal in the SE sequence. This cross-linking could spatially couple different phase behaviors within a condensate, where the photoresponsive DNA phase, undergoing an increase/decrease in the fluidity, hydrodynamically performs mechanical actions upon the non-photoresponsive DNA phase. Notably, we discovered multiple modes in the photogenerated mechanical actions as a function of the applied temperature and Azo insertion site in the SE. The modes observed were: (1) diffusion-driven outward-directed spread under UV irradiation, (2) elastocapillarity effects capable of collapsing sparsely arranged gel structures into smaller sizes under UV, and (3) cycle flows of alternate UV-generated spread and Vis-induced convergence capable of collecting the scattered DNA gel particles. With quantitative experimental validation of the photoswitched SEs binding stability, we elucidate the thermodynamic origin of the multi-modal mechanical actions by DNA microflow that depends on the Azo insertion site. Lastly, we demonstrate the directional motion of a photoresponsive DNA liquid condensate by localizing the irradiated area within a fraction of the condensate. Localized UV–Vis photoswitching produced two types of transient yet distinct swimming behaviors of the condensate, with the direction, lifetime, and morphological changes depending on the switching frequency. At lower switching frequencies, the DNA liquid condensate can swim like a jellyfish, and at higher frequencies, the condensate can be pulled by its localized cycle flow. By modeling mass transport and accompanying momentum exchange between the photoirradiated and non-irradiated regions within a liquid DNA condensate, we reveal a regulatory role of the switching frequency in the energy transport between the two regions.

## Results

### Photoresponsive DNA condensates

DNA condensates capable of photoresponsive state transition are achieved by introducing Azo, a photoisomerizable chemical compound, into nano-scale Y-shaped motifs of DNA (Y-motifs, Fig. 1a). The Y-motifs (Supplementary Tables 1–4) had three-branched double-stranded stems that terminated in an overhang of a single-stranded SE with eight nucleotides (nt). The self-complementarity of the SEs provides the inter-motif connectivity, which allows the nanoscale Y-motifs to self-assemble into micro-scale DNA condensates. A prescribed annealing process caused a set of three ssDNAs to self-assemble into a Y-motif through the stem hybridization, subsequently forming micro-scale condensates through a duplex formation of eight-base-pair (bp) SEs. One or two Azo residues were inserted into the SEs to provide the motifs with photoresponsive capabilities. Azo undergoes trans-to-cis or cis-to-trans photoisomerization upon UV (300 nm < λ < 400 nm) or Vis (λ > 400 nm) irradiation, respectively. cis-Azo destabilizes the base stacking of hybridized DNA strands and causes the hybridized strands to dissociate due to steric hindrance, whereas trans-Azo strengthens the duplex formation by stabilizing the base stacking[65]. Consequently, the Azo-modified DNA nanostructures can be reversibly dissociated/associated with each other in response to UV or Vis irradiation, respectively, due to the photoreversible cis/trans isomerization. The SEs photoswitching provides the principal mechanism for the photoregulated state transition of DNA condensates (Fig. 1b).

First, we demonstrate a UV-induced gel-to-liquid state transition of an Azo-modified DNA gel particle at a constant temperature of 46°C (Fig. 2a. Compare (i) with controls (ii)/(iii). See Supplementary Table 3 for the sequence design). The constructed DNA condensates were

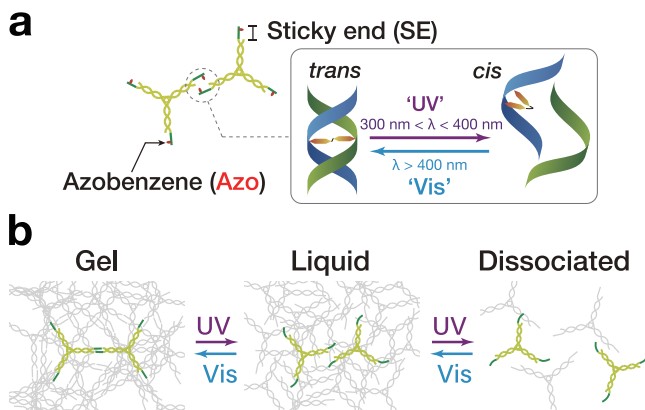

**Fig. 1 | Photocontrolled state transition of DNA condensates.** Schematics of **a** Y-shaped nanostructures of DNA (Y-motifs) with azobenzene (Azo) inserted in the sticky ends (SEs) and **b** photoswitchable state transition of DNA condensates.

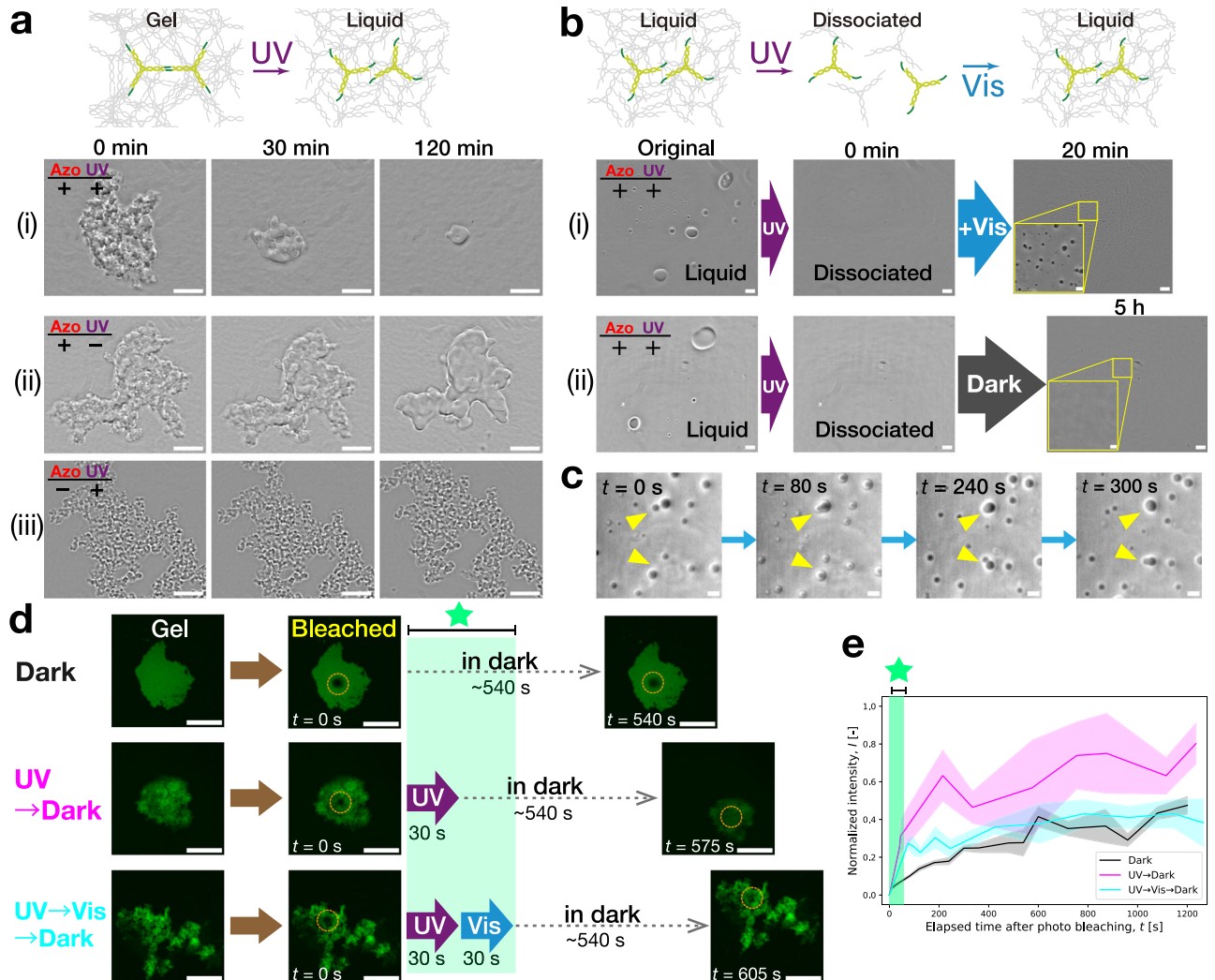

**Fig. 2 | Photoreversible phase-state control of Azo-modified DNA condensates at constant temperatures.** Time-sequential images acquired by phase contrast (PC) microscopy of **a** a gel-to-liquid phase change under UV at 46°C and **b** a UV-induced liquid-to-dissociated state change ('Original' to '0 min') reversed by additional Vis irradiation at 53°C. **a** (ii), (iii), **b** (ii): controls. **c** Coalescence of the photoreversed DNA liquid condensates captured in (**b**). Time-sequence started 52 min after switching to Vis irradiation. Yellow triangles mark coalescing condensates. **d** Procedures and conditions of **e** fluorescence recovery after photobleaching (FRAP) of Azo-modified DNA gel-state condensates at 35°C in a dark environment ('Dark'), UV irradiation followed by the dark ('UV→Dark'), and consecutive UV and Vis followed by the dark ('UV→Vis→Dark'). Each solid line connects the mean values at specific elapsed times for clarity. Shaded areas indicate S.D. ($n \geq 4$). See Supplementary Fig. 3d for the individual data points. **d, e** Shaded rectangles in pale green highlight the post-bleaching irradiation conditions. Irradiation intensities: (**a**) 1.1 mW cm$^{-2}$ in the first 60 min of UV, 0.60 mW cm$^{-2}$ in the additional irradiation; (**b, c, e**) 8.9 mW cm$^{-2}$ (UV), 10.4 mW cm$^{-2}$ (Vis). Scale bars: (**a, b, d**) 50 μm; (**b** inset) 10 μm; (**c**) 20 μm. Source data are provided as Source Data files.

irradiated for 2 h with sufficiently weak UV to capture their morphological changes while minimizing disturbance flows around the condensates. We used an imaging interval of 30 min to minimize the exposure of the samples to visible bright-field light during UV irradiation. Consecutive phase-contrast (PC) images show that the sponge-like surface smoothed out within 30 min of UV irradiation (365 nm), accompanied by a large rounding-up of the structure within 120 min. The significant changes in the size and morphology (Fig. 2a) indicate that the condensates achieved a significant gel-to-liquid phase transition within ~10 min of UV irradiation. The UV-induced rounding-up of condensates strongly suggests an increase in their fluidity as a consequence of the minimization of a surface-to-volume ratio in the gel-to-liquid phase-transitioning condensates (rheological discussions are available in Supplementary Texts and Supplementary Fig. 7a)[66-68]. The slight surface smoothening observed in the control experiment using Azo in the absence of UV irradiation (Fig. 2a(ii)) indicates that Azo insertion by itself can affect the SEs' thermostability, as mentioned later.

Additionally, time-lapse confocal imaging of UV-irradiated gel-state condensates at 56°C also shows a viscous threading and rounding-up of the irregularly shaped condensates (Supplementary Movie 1), suggesting increased fluidity. Finally, the liquefied DNA condensates dissociated after further UV irradiation.

Next, Fig. 2b shows a Vis-induced dissociated-to-liquid transition at a constant temperature of 53°C (compare (i) with (ii)). Initially, DNA condensates were dissociated in UV irradiation at this temperature. Upon switching to Vis (440 nm), noticeable nucleation occurred within 1 min, followed by significant particle size growth within 20 min (Supplementary Fig. 2a, b). This indicates that the SE reassociation due to *cis*-to-*trans* Azo isomerization led to the reconstruction of DNA condensates without annealing procedures (see also Supplementary Movie 2 for time-lapse confocal microscopy imaging of Vis-induced recondensation at 56°C). We note that the Vis-recondensed DNA appeared in smaller sizes than the initial sizes prior to UV irradiation (Fig. 2b(i). Compare '20 min' with 'Original'). This inefficiency in the Vis-reversed process is discussed by obtaining the size distributions of

recondensed DNA particles as a function of Vis irradiation time (Supplementary Fig. 2).

To scrutinize the effects of thermal *cis*-to-*trans* isomerization of Azo molecules[69] on the recondensation, we performed a control experiment under dark conditions over an extended period. Even after 5-h Vis irradiation, no distinct condensate growth was observed (Fig. 2b(ii)), suggesting that the thermal *cis*-to-*trans* Azo isomerization contributed negligibly to the recondensation process.

Further, time-lapse observation of the equilibrated state (Fig. 2c) captured the coalescence of contacting DNA particles, suggesting that the reconstructed DNA condensates equilibrated in a liquid state[24,49]. Recondensation was observed only at temperatures exceeding 50°C, indicating that frequent interactions between SEs, owing to substantial thermal fluctuations, are essential for the recondensation. Overall, we verified that the photoresponsive DNA condensates can modulate their fluidity through reversible state transitions.

To confirm the molecular-level basis for the photoreversible SE binding, we carried out native polyacrylamide gel electrophoresis (PAGE). Here, we used Y-motifs with a single SE to highlight the SE binding states while preventing condensation. As illustrated in Supplementary Fig. 4a, the single-SE motifs adopt only two states, the associated (dimer) and dissociated (monomer) states, corresponding to the upper and lower bands in a lane, respectively. The samples to be loaded in the lanes were irradiated by 6-min UV, followed by 6-min Vis. With UV irradiation, we confirmed an increased fraction of the monomer state (unbound SEs), while switching to Vis led to a recovery of the original fraction (compare the data in orange and green boxes in Supplementary Fig. 4b, c). The PAGE results suggest that the observed photoinduced phase changes (Fig. 2a, b) resulted from the SE photoreversible binding/unbinding, rather than irreversible UV-induced structural breakdown of the motif structures.

The irradiation-induced fluidity alterations (Fig. 2a, b) were indicated as a consequence of the phase behaviors of DNA condensates. Motif mobility in the condensate is a source of the fluidity in supramolecular self-assemblies[70]. Thus, we further provide experimental support for the diffusion properties of the constituting motifs in DNA condensates composed of triple-SE motifs, using fluorescence recovery after photobleaching (FRAP, Fig. 2d, e). In each FRAP experiment, a DNA gel particle (*trans*-Azo) was photobleached within a 6-μm spot before 30 s of UV/Vis irradiation upon the entire region. To prevent photobleaching from affecting the condensate phase state, we preferred the bleaching to precede the irradiation step (Fig. 2d), which was opposite to a typical FRAP procedure, where specific reactions come before the bleaching step (see Supplementary Texts and Supplementary Fig. 3 for the detailed description of the protocol and analysis method). Further, we chose a working temperature of 35°C to stabilize the resulting condensate in a gel state. At higher temperatures, which enhance liquid-state condensation, UV–Vis differences in the diffusion properties would become less distinct. The FRAP results showed a greater increase in the fluorescence recovery in UV irradiation than in no-irradiation conditions (Fig. 2e. Compare 'UV→Dark' with 'Dark'. See Supplementary Fig. 3d for the individual data points). In the alternating UV and Vis irradiations, a well-enhanced recovery was observed within the first 30 s of UV irradiation (compare 'UV→Vis→Dark' with 'UV'), which was followed by a decrease to the no-irradiation level after switching to Vis irradiation (compare 'UV→Vis→Dark' with 'Dark'). These FRAP results confirm the photoreversibility in the motif mobility in the condensates.

Taken together, these data provide experimental support that the photogenerated SE binding state and consequent motif mobility determine the phase state and fluidity of DNA condensates.

## Evaluation of SE binding strength

Above, Azo insertion site was not highlighted in the demonstration of the photoreversible phase-state change of DNA condensates. To evaluate the effect of the insertion site on the SE binding strength, we measured the dissolving temperature $T_D$ of DNA condensates as an indicator of the thermostability. $T_D$ is defined as a temperature above which DNA condensates are stably dissociated in the buffer (Fig. 3a). To measure $T_D$ for the *cis*/*trans* states, DNA condensates were exposed to UV/Vis irradiation and then heated until complete dissolution. We considered triple-SE Y-motifs with different Azo insertion sites in the SE (Fig. 3b). Throughout, the Azo-containing SEs are referred to as $SE_{1x7}$, $SE_{3x5}$, and $SE_{2x1x5}$, respectively, where the subscript specifies the Azo insertion site 'x' in the 8-nt SE. For example, in $SE_{1x7}$, Azo residue divides the first 1 nt from the 5' end and the remaining 7 nt in the SE. Hereafter, Y-motif with each branch terminating in $SE_i$ ($i$ = 1x7, 3x5, 2x1x5) will be referred to as $Y_i$ ($i$ = 1x7, 3x5, 2x1x5). We found that while $Y_{1x7}$-based DNA condensates showed the highest $T_D$, or the strongest binding stability, among the photoresponsive SEs for both *cis*/*trans*, followed by $Y_{3x5}$ and then $Y_{2x1x5}$ (Fig. 3b). Furthermore, the *cis*–*trans* gap of $T_D$ showed an increase in this order. Further, we noted that Azo insertion in the SEs by itself can lower $T_D$ of the condensates (compare 'Control' with '$SE_{3x5}$' and '$SE_{2x1x5}$' in Fig. 3b), which may be responsible for the slight surface smoothening in the control experiment using $Y_{3x5}$ in the absence UV irradiation (Fig. 2a(ii)).

The dependence of SE binding strength on the Azo insertion site was also confirmed by PAGE experiments. For varied insertion sites in the SE, samples of single-SE Y-motifs were exposed to UV (0–6 min) and Vis (additional 6 min) irradiation (Supplementary Fig. 4). The concentration of the binary states in each lane was quantified as a function of the irradiation conditions. We observed that the amplitude in the UV–Vis band intensity shift varied as a function of the Azo insertion site in the SE (Supplementary Fig. 4c): in UV-to-Vis photoswitching, $SE_{2x1x5}$ showed the most pronounced change, $SE_{3x5}$ a moderate shift, and $SE_{1x7}$ the lowest amplitude. This Azo insertion site-dependence of the concentration of the binary states agreed well with the widening gap of $T_D$ between *trans* and *cis* states (Fig. 3b).

These results revealed that the location and number of Azo residues in the SE significantly affected the binding strength of the photoresponsive DNA motifs. An inward shift of Azo from 5' to 3' end in the SE (from $SE_{1x7}$ to $SE_{3x5}$) destabilized the binding strength, and an increase in the number (from $SE_{3x5}$ to $SE_{2x1x5}$) had a further destabilizing effect. At the molecular level, the bound and unbound states of SEs switch interchangeably from one state to the other in chemical equilibrium. In $Y_{3x5}$, where the two opposing Azo residues are closely located near the middle region of base-paired SEs, there is the possibility that the close residues may fall into *cis* coincidentally, leading to enhanced steric hindrance that disfavors the hybridized state. Similarly, in $Y_{2x1x5}$, two sets of Azos dominate the middle region of the hybridized SEs, causing additional instability in the SE binding. Conversely, in $Y_{1x7}$, the inserted Azo residues located near the terminal of the base-paired SEs are distanced apart from each other, thus causing a lesser possibility of the simultaneous steric hindrance.

## Programmed mechanical actions using microflow of DNA condensates

**Multi-mode photogenerated microflow of DNA condensates.** We designed DNA motifs combining Azo-containing and Azo-free Y-motifs, aided by DNA sequence programmability, to create mechanical actions. One of the combined motifs was photoresponsive $Y_i$ ($i$ = 1x7, 3x5, 2x1x5, green in Fig. 4a), and the other was non-photoresponsive Y-motif (referred to as $Y_0$, blue in Fig. 4a) that was orthogonal to $Y_i$ in the SE sequence. In a liquid state, these DNA condensates would not coalesce due to their molecular recognition capability[24]. Introducing six-branched cross-linker DNA $L_0$, three of whose six SEs selectively bonded with $Y_i$ and the other three with $Y_0$ (Fig. 4b)[24,29], enabled the formation of an adhered structure of immiscible DNA condensates. Hereafter, the cross-linked DNA motifs forming the adhered DNA condensates will be referred to as $Y_i/L_0/Y_0$. Sequence-specific

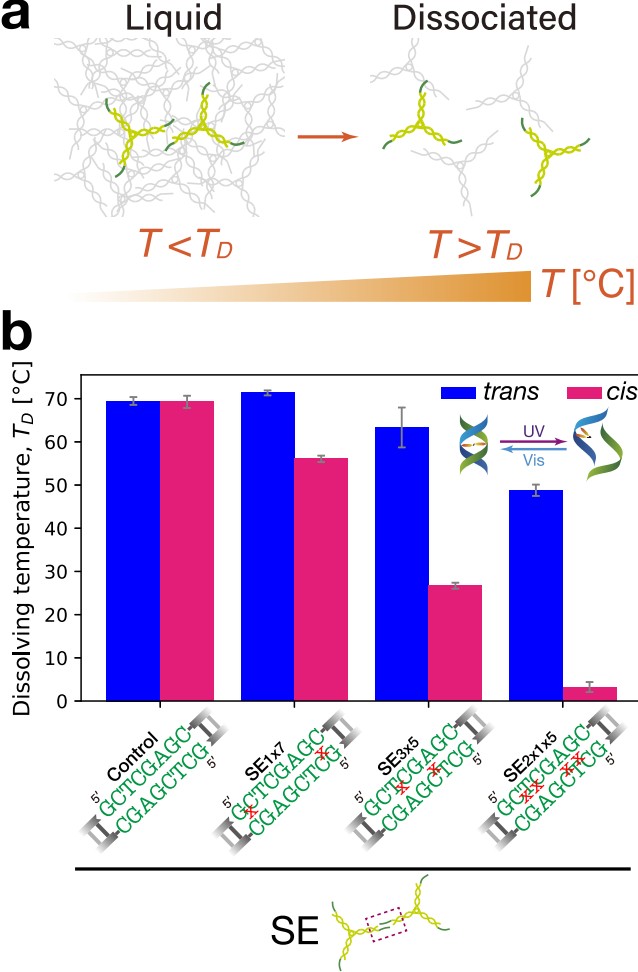

**Fig. 3 | Experimental evaluation of SEs' binding strength. a** Dissolving temperature $T_D$ as a quantifiable indicator of the binding strength. Above $T_D$, DNA condensates become dissociated with increasing temperature $T$. **b** Bar plots of $T_D$ as a function of the insertion site of Azo in the SE. Error bars: S.D. ($n = 3$). Note that for the control, the blue and magenta bars refer to the Vis and UV irradiation instead of *trans* and *cis*, respectively. For *trans* and *cis*, DNA condensates were irradiated by Vis (4.4 mW cm⁻²) for 5 min and UV (0.34 mW cm⁻²) for 10 min, respectively. Source data are provided as Source Data files.

photoresponsive capabilities were realized in the adhered DNA condensates due to the introduction of Azo in the SEs of $Y_i$ (Fig. 4c, Supplementary Table 10). Various adhered DNA condensates in a gel state (Supplementary Fig. 5) showed that the cross-linker DNA could spatially couple different photoinduced phase behaviors within one system.

Another advantage of the cross-linked motif design was the traceability of the photoresponsive DNA microflow in the quantification of its mobility, as shown below. The cross-linking and consequent high affinity between $Y_i$ and $Y_0$ allowed the $Y_0$ gel particles to serve as a good tracer of the $Y_i$ microflow motion. Commercially available microbeads, which would not favor to be fully embedded within the condensate, were disqualified for the tracing assays.

Due to the coexistence of sequence-specifically photoresponsive DNAs, energy-transducing systems (Fig. 5a) were formed, where the photoresponsive DNA condensates (Fig. 5a, green) received light as an energy source and an on/off switching signal. The constructed DNA condensates of cross-linked motifs were irradiated with collimated excitation light ejected from an objective lens of the microscope with sufficient intensity for the drastic generation of microflows. These irradiation intensities were higher than those used for the basic

characterization of the photoinduced phase change, with an emphasis placed on the minimization of disturbance flows (Fig. 2). The intensified irradiation strength significantly reduced the reaction timescales from minutes to seconds. The resulting photoinduced state transition generated a microflow acting on the non-photoresponsive DNA condensates (Fig. 5a, blue). The photogenerated mechanical actions of photoresponsive $Y_i$ DNA exerted on non-photoresponsive $Y_0$ DNA can be categorized into three modes, based on their flow direction and applied irradiation (see the summary in Fig. 5b). It is reminded that the initial states shown in Fig. 5 were selected specifically as configurations well suited to clearly display the distinctive flow modes. See other representative initial configurations in Supplementary Fig. 5.

The first mode is "spread" mode (Fig. 5c), which was observed for $Y_{2\times1\times5}/L_0/Y_0$ motifs (< 40 °C), $Y_{3\times5}/L_0/Y_0$ (45–60 °C), and $Y_{1\times7}/L_0/Y_0$ (> 60 °C). In this mode, $Y_i$ DNA dissolved rapidly into the buffer due to a fast gel-to-dissociated state transition upon UV irradiation, leading to an outward spread of $Y_0$ DNA particles (Fig. 5d, Supplementary Movie 3).

Another mode is "collapse" mode (Fig. 5e), which was observed only for $Y_{1\times7}/L_0/Y_0$ (35–50 °C). Figure 5f shows a sparsely arranged structure of the adhered DNAs in the initial state. Upon UV irradiation, the $Y_{1\times7}/L_0/Y_0$ condensates showed a substantial compaction and even folding into a smaller size (Fig. 5f, Supplementary Movie 4). The compaction and self-folding process can be explained as a wetting process of the gel-to-liquid phase-transitioning DNA phase ($Y_{1\times7}$ condensates) interfaced with the deformable DNA gel phase ($Y_0$ condensates), as discussed in Supplementary Texts (Supplementary Fig. 7b)[10,67,68,71].

The last mode is "spread and collect" mode (Fig. 5g), which was observed for $Y_{2\times1\times5}/L_0/Y_0$ (40–45 °C) and $Y_{3\times5}/L_0/Y_0$ (55–60 °C). At the initial state, the immiscible $Y_i$ and $Y_0$ DNA condensates constituted a compartmentalized condensate, wherein the $Y_i$ condensates encompassed the $Y_0$ condensates. Upon UV irradiation, the $Y_i$ DNA favored the liquid-to-dissociated state transition, leading to a spread of the $Y_0$ DNA condensate (Fig. 5h(i)). Upon switching to Vis (520–550 nm) irradiation, the diffused-out $Y_i$ DNA showed a reversing flow due to a dissociated-to-liquid state transition, while dragging the $Y_0$ DNA condensates backward, finally leading to the reassembly of a compartmentalized condensate as the initial state (Fig. 5h(ii), Supplementary Movie 5). The observed reversing flow suggests that the Vis-induced reassociation of the SEs increased entropic elasticity, which served as an attractive force to reverse the flow and collect the $Y_0$ DNA. The forward and reverse cycles, resembling muscle actions, could be repeated in several cycles at 55 °C (Supplementary Fig. 6, Supplementary Movie 6). This limitation shows the imperfect photoswitching efficiency of the Azo molecule and also the bulky geometry of the observation chamber used (for a more detailed discussion, see Supplementary Texts[69,72–74]).

Figure 5b shows the summary of the flow modes observed for each Azo insertion and their effective temperature ranges, in connection with the dissolving temperature $T_D$ from Fig. 3. With an increase in $T_D$, the effective temperature range of each flow mode increased to a higher range. The collapse mode was only available for the lowest *cis–trans* $T_D$ gap in $Y_{1\times7}/L_0/Y_0$. The spread and collect cycles were observed at temperatures of 40–45 °C ($Y_{2\times1\times5}/L_0/Y_0$) and 55–60 °C ($Y_{3\times5}/L_0/Y_0$), which were close to the dissolving temperature $T_D$ of the corresponding photoresponsive DNA condensates (*trans*, Fig. 3b), 50 °C ($Y_{2\times1\times5}$) and 65 °C ($Y_{3\times5}$), respectively.

**Temperature dependence of flow mobility.** Each flow mode exhibited a greater mobility in the moderate temperature range than in the lower and higher temperature ranges (*e.g.*, the spread mode of $Y_{2\times1\times5}/L_0/Y_0$ in Fig. 6a and the collapse mode of $Y_{1\times7}/L_0/Y_0$ in Fig. 6b). To quantitatively analyze this temperature-dependent flow mobility, we derived the mean square displacement (MSD) of $Y_0$ DNA particles

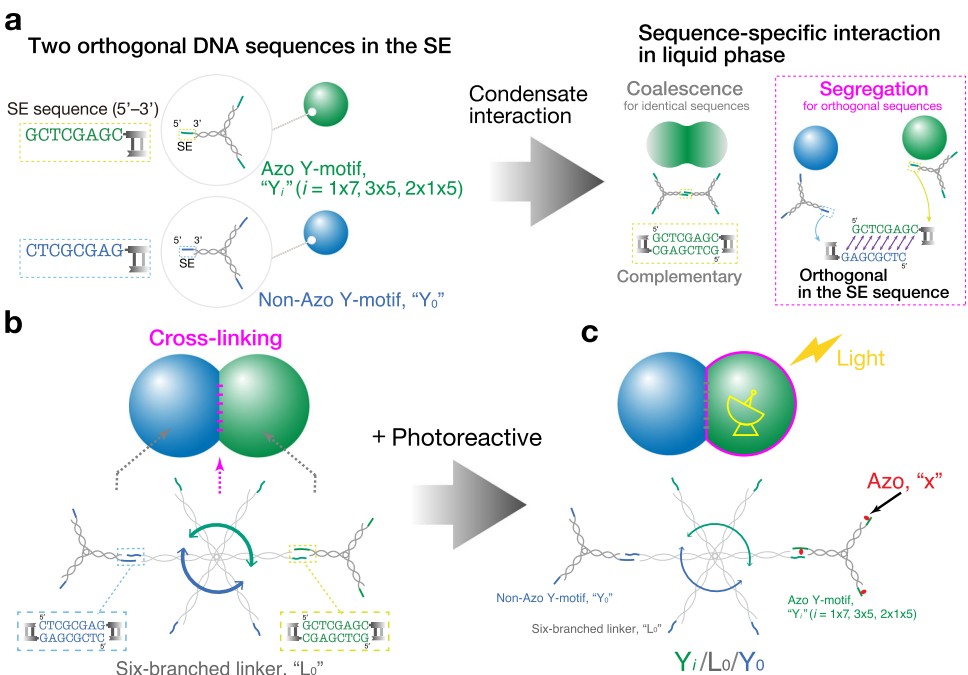

**Fig. 4 | Sequence-specific photoresponsive capability. a** Sequence-specific selectivity in condensate interaction. (Left) Two orthogonal SE sequences. Photo-responsive Y-motif with Azo-bearing SE ($SE_i$) is referred to as $Y_i$ ($i = 1x7, 3x5, 2x1x5$); non-photoresponsive Y-motif without Azo as $Y_0$. (Right) Due to the palindromic sequence in the SE, the resulting DNA liquid condensates with the same SE sequence favor coalescence; those with orthogonal sequences in the SE favor segregation. **b** A six-branched cross-linker $L_0$ that binds to orthogonal $Y_i$ and $Y_0$ forms an adhered structure of the two immiscible DNA condensates. Three neighboring SEs (blue and green) in $L_0$ exclusively hybridize with $Y_0$ and $Y_i$ motifs, respectively. **c** Sequence-specific photoresponsive capability by introducing Azo in one of the cross-linked orthogonal motifs. Below each image, $Y_i/L_0/Y_0$ denotes the corresponding cross-linked DNA motifs containing $Y_i$ ($i = 1x7, 3x5, 2x1x5$, green, FAM). Scale bars: 20 μm.

serving as a tracer in $Y_i/L_0/Y_0$ ($i = 1x7, 3x5, 2x1x5$) condensate micro-flows. A range of temperatures from RT to 60°C ($i = 1x7, 3x5$) and 45°C ($i = 2x1x5$) was investigated. The resultant trajectories of $Y_0$ particles were subjected to particle tracking analysis using TrackMate[75], an open-source Fiji plugin (Supplementary Fig. 8), and subsequent conversion to MSD plots. We provide representative MSDs as a function of time interval $\tau$ for the spread mode of $Y_{2x1x5}/L_0/Y_0$ (Fig. 6c) and the collapse mode of $Y_{1x7}/L_0/Y_0$ (Fig. 6d) from low, middle, and high temperature ranges. We confirmed that the MSD curves showed marked upward curvature in the middle temperature ranges, compared to those in the lower and higher temperature ranges. The marked upward curves in the middle temperature range indicate the ballistic nature of the photogenerated migrations rather than the random Brownian motion, which normally displays a linear slope in the MSD curve (Supplementary Texts)[76,77].

For a comprehensive evaluation of the flow mobility at various temperatures, we further calculated the diffusion coefficients $D'$ of individual trajectories of $Y_0$ particles from linear curve fitting against their MSD curves (Supplementary Eq. (2)). To consider the direction of the generated migrations, we arbitrarily defined a signed diffusion coefficient $D^*$, where $D^* = -D'$ in the case of inward-directed flow, and $D^* = D'$ otherwise (Fig. 6e). The signed $D^*$ was plotted as a function of $T$ for alternating UV and Vis irradiation (Fig. 6f).

In Fig. 6f, we found distinct single-peak profiles in the "spread" mode for $Y_{2x1x5}/L_0/Y_0$ and $Y_{3x5}/L_0/Y_0$ and the "collapse" mode for $Y_{1x7}/L_0/Y_0$ ('UV' panel) and the reverse phase of the "spread and collect" mode for $Y_{2x1x5}/L_0/Y_0$ and $Y_{3x5}/L_0/Y_0$ ('+Vis' panel). Interestingly, we found a marked peak shift by 20°C between $Y_{3x5}/L_0/Y_0$ and $Y_{2x1x5}/L_0/Y_0$ and a drastic peak inversion between $Y_{3x5}/L_0/Y_0$ and $Y_{1x7}/L_0/Y_0$. On a closer inspection, $Y_{1x7}/L_0/Y_0$ showed a negative-to-positive sign inversion close to 60°C, indicating a mode shift from the "collapse" to "spread" modes. These drastic $D^*$ profile changes show that the microflow of DNA condensates can undergo a drastic mode switch via

one- or two-base relocation of Azo residues in the SE without re-designing the motif sequences.

**Azo insertion site determines the enthalpy difference in SE binding/unbinding.** Thermodynamically, the observed peak shift (between $Y_{3x5}/L_0/Y_0$ and $Y_{2x1x5}/L_0/Y_0$) and sign inversion (between $Y_{3x5}/L_0/Y_0$ and $Y_{1x7}/L_0/Y_0$) in the temperature dependence of flow mobility $D^*$ (Fig. 6f) can be associated with the enthalpy difference in the SEs (Supplementary Fig. 9). This corresponds to a difference between the enthalpy of the bound state of SEs and that of the unbound state of the SEs at equilibrium (Supplementary Fig. 10a). Irradiation on DNA con-densates leads to Azo isomerization in the SEs, accompanied by a dynamic shift between $\Delta H_{trans}$ and $\Delta H_{cis}$; that is, $|\Delta H|$ corresponds to the binding strength of the SEs. Here, we introduce a phase diagram that spans the $|\Delta H| - T$ space (Supplementary Fig. 10b). In this phase map, an irradiation-induced *trans–cis* Azo isomerization in the SEs constitutes a vertical reaction pathway of a length $|\Delta(\Delta H)_{trans \to cis}|$ at a specific temperature. The location and length of the reaction pathway determine the peak location and sign of a $D^*$ profile in the $T$ dependence of flow mobility. Further, with experimental evaluation of various Azo-free SEs (Supplementary Fig. 10c. Their sequence designs are listed in Supplementary Tables 11–17), we made quantitative estima-tions of $|\Delta H_{trans}|$ and $|\Delta H_{cis}|$ in the Azo-modified SEs ($SE_{1x7}$, $SE_{3x5}$, and $SE_{2x1x5}$), and found an increase in $|\Delta(\Delta H)_{trans \to cis}|$ in the order of $SE_{1x7}$, $SE_{3x5}$, and $SE_{2x1x5}$ (Supplementary Fig. 10d). In the phase map, these estimated $|\Delta(\Delta H)_{trans \to cis}|$ values were consistent with the observed flow mobilities and signs of $D^*$ (Supplementary Fig. 10e). More in-depth thermodynamic discussion based on the quantification of $|\Delta(\Delta H)_{trans \to cis}|$ values is available in Supplementary Texts.

**Directional motion of DNA liquid condensates driven by localized photoswitching.** The irradiated area used above was radially sym-metric, resulting in no net directional displacement of the liquid

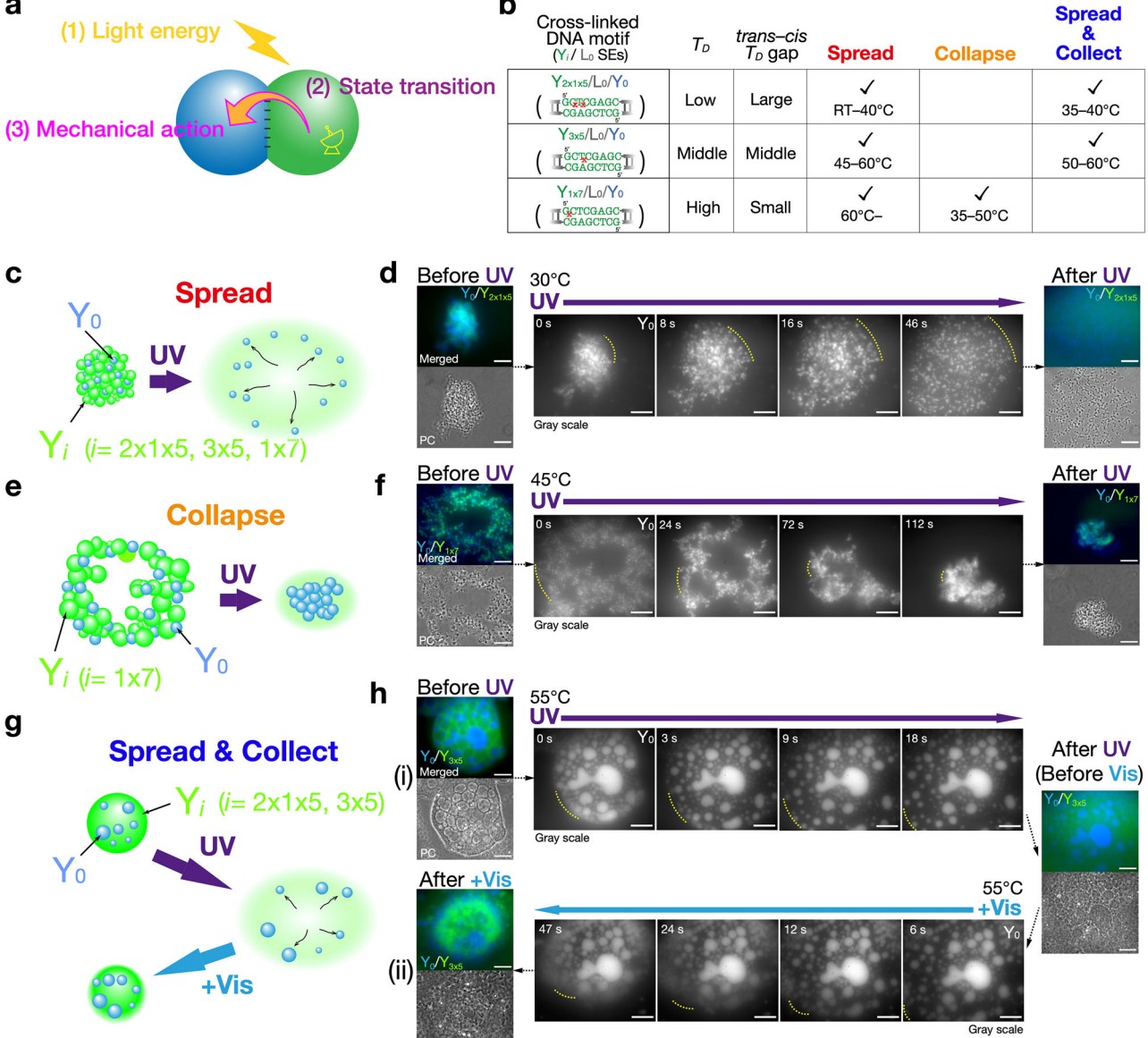

**Fig. 5 | Mechanical actions using microflow of DNA condensates. a** Schematic of mechanical action of state-transitioning DNA upon non-photoresponsive DNA. **b** Summary table of (**c**, **e**, **g**) generated flow modes and their observed temperature ranges for different insertion sites of Azo in the SE. Dissolving temperature $T_D$ for the corresponding SEs (Fig. 3) are also related. **c**, **d** "Spread" mode due to UV-induced fast diffusion. **c** Upon UV irradiation, photoresponsive ($Y_i$) DNA favors a fast gel-to-dissociated state transition, leading to a fast spread of non-photoresponsive $Y_0$ DNA in the outward direc*t*ion. $i = 2x1x5, 3x5, 1x7$. **d** (Top) Merged fluorescence and (bottom) PC images of a $Y_{2x1x5}/L_0/Y_0$ condensate captured (leftmost) before and (rightmost) after UV irradiation at 30°C. (Middle) Time-sequential fluorescence images of spreading $Y_0$ particles in response to UV irradiation. **e**, **f** "Collapse" mode due to a UV-induced gel-to-liquid state transition. **e** Upon UV irradiation, $Y_i$ condensate shows a gel-to-liquid state transition, resulting in a collapse of $Y_0$ gel particles. $i = 1x7$. **f** Merged fluorescence and PC images of a $Y_{1x7}/L_0/Y_0$ condensate captured (leftmost) before and (rightmost) after UV irradiation at 45°C. (Middle) Time-sequential fluorescence images of collapsing $Y_0$ particles in UV irradiation. **g**, **h** "Spread and collect" mode in alternating UV/Vis irradiation. **g** Initially, $Y_i$ and $Y_0$ condensates coexist in a condensate as a liquid state. Upon UV irradiation, a $Y_i$ liquid condensate shows "spread" behavior due to a liquid-to-dissociated state transition and releases $Y_0$ condensates; upon switching to Vis irradiation, the dissociated $Y_i$ shows reversing flow due to Vis-induced reassociation of SEs, and collects the released $Y_0$ condensates. **h** Merged fluorescence and PC images of a $Y_{3x5}/L_0/Y_0$ condensate captured (top leftmost) before UV, (rightmost) after UV, and (bottom leftmost) after additional Vis irradiation at 55°C. Time-sequential fluorescence images of (top middle, left-to-right) spreading $Y_0$ condensates under UV irradiation and (bottom, right-to-left) their collection after switching to Vis. Yellow dashed-line curves mark the advancing periphery of the condensate as an eye guide. Irradiation intensities: (UV, 360–370 nm) 3.3 mW cm$^{-2}$; (Vis) 7.0 mW cm$^{-2}$ (460–495 nm for FAM), 4.8 mW cm$^{-2}$ (520–550 nm for Cy3). Scale bars: 50 μm.

condensates. To achieve the directional motion of a DNA liquid condensate, we localized the switching irradiation in a specific region smaller than the condensate, using a digital mirror device (DMD). The irradiated area was specified as a region-of-interest (ROI) using the driver software (Fig. 7a). At each photoswitching, the ROI was repositioned stepwise to overlie the liquid-state migrating condensate (in UV irradiation) and the UV-dissociated phase (in Vis irradiation). Here, the

SE length was reduced from 8 nt to 6 nt to lower the working temperature ($Y_{2x4}$, Supplementary Table 5), as explained later. A wide range of switching frequencies ('low', 'intermediate', and 'high') was attempted manually.

At low switching frequencies, localized UV irradiation upon a DNA condensate created a spreading DNA flow from the ROI, from which the condensate surface slowly showed large unfurling. The condensate

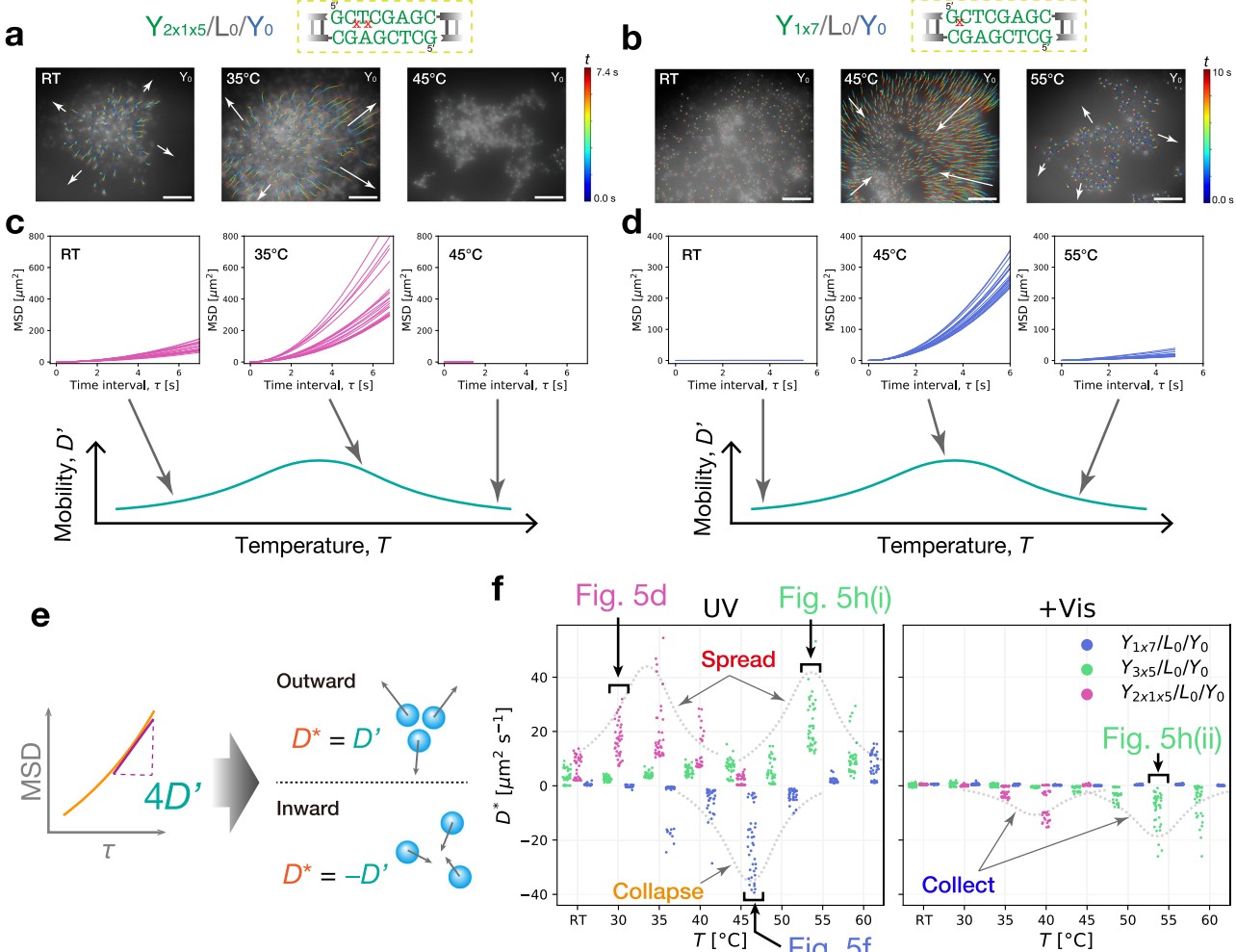

**Fig. 6 | Temperature-dependent flow mobility of mode-switchable photo-induced microflow of DNA condensates.** Particle tracking of non-photoresponsive $Y_0$ DNA particles for **a** "spread" mode of $Y_{2x1x5}/L_0/Y_0$ and for **b** "collapse" mode of $Y_{1x7}/L_0/Y_0$. Detected trajectory paths are visualized using a color map of an elapsed time $t$. Arrows indicate the flow directions. Representative plots of calculated mean square displacements (MSDs) as a function of time interval $\tau$ for the **c** "spread" mode (from **a**) and for the **d** "collapse" mode (from **b**). **e** Signed diffusion coefficient $D^*$, arbitrarily defined to allow for the direction of flow. Diffusion coefficient $D'$ expressing the mobility of individual migration paths was calculated by taking the linear curve fitting against a calculated MSD curve. $D^*$ was then determined by adding a positive or negative sign to $D'$ depending on the direction of the generated flow. **f** Plots of signed $D^*$ as a function of temperature $T$ in UV and following Vis irradiation. Data points of $Y_i/L_0/Y_0$ ($i = 1x7, 3x5$) are staggered sideways for clarity at each temperature examined. The flow modes presented in Fig. 5d, h are related. Scale bars: 50 μm. Source data are provided as Source Data files.

was then pushed away from the ROI until an equilibrium. Switching to Vis reversed the UV-diffused DNA to the pushed condensate, allowing for the next cycle. Multiple slow cycles of the condensate unfurling and closing at the aft end continued until its complete crumble. This swimming behavior is referred to as 'push-swimming'. Its morphological dynamics resembled that of a swimming jellyfish (Fig. 7c, Supplementary Movie 7). Transient yet distinct push-swimming was observed in more than 8 out of 11 consecutive attempts (Fig. 7b).

At higher but intermediate switching frequencies, UV-discharged DNA microflow from a condensate was reversed to the ROI without the onset of distinct propulsion (Fig. 7d). Interestingly, unlike push-swimming, this reversed flow pulled the condensate toward the ROI over a finite distance (Fig. 7f, Supplementary Fig. 11c, Supplementary Movie 8). The rapid switching iterations resulted in alternate opening and closing of the migrating condensate at the fore end. This behavior, which we term 'pull-swimming', was observed in more than 7 out of 10 consecutive trials (Fig. 7e). Compared to push-swimming, pull-swimming exhibited smaller stepwise displacements (Fig. 7h, Supplementary Fig. 11c) but had a longer lifetime, indicated by a plateau in the time evolution of displacements (Fig. 7h), due to the limited diffusion in frequent switching.

At much higher switching frequencies, we only observed minimum displacements, where no DNA microflow discharged from the condensate.

Further, we employed these two types of condensate swimming for cargo transport to exemplify their great potential as a microfluidic tool in a broad range of applications. The cargo loading was achieved using cross-linked DNA motif $Y_{2x4}/L'_0/Y_0$ that combined orthogonal $Y_{2x4}$ and $Y_0$ (Supplementary Table 18). The resulting cargo-loaded DNA condensates could encompass $Y_0$ gel particles as cargo at the working temperatures (Fig. 7i, j). As mentioned above, the SE length was reduced from 8 nt to 6 nt to lower the working temperature. We preferred the gel-state cargo to minimize the coalescence and growth of cargo particles, which was a significant impedance to cargo transport.

In Fig. 7i, the push-swimming of a cargo-loaded condensate trailed cargo particles, narrowly tethered to its aft end, over a finite distance (Supplementary Movie 9). This limited cargo transport distance (~10 μm) was shorter than that of the transporting condensate

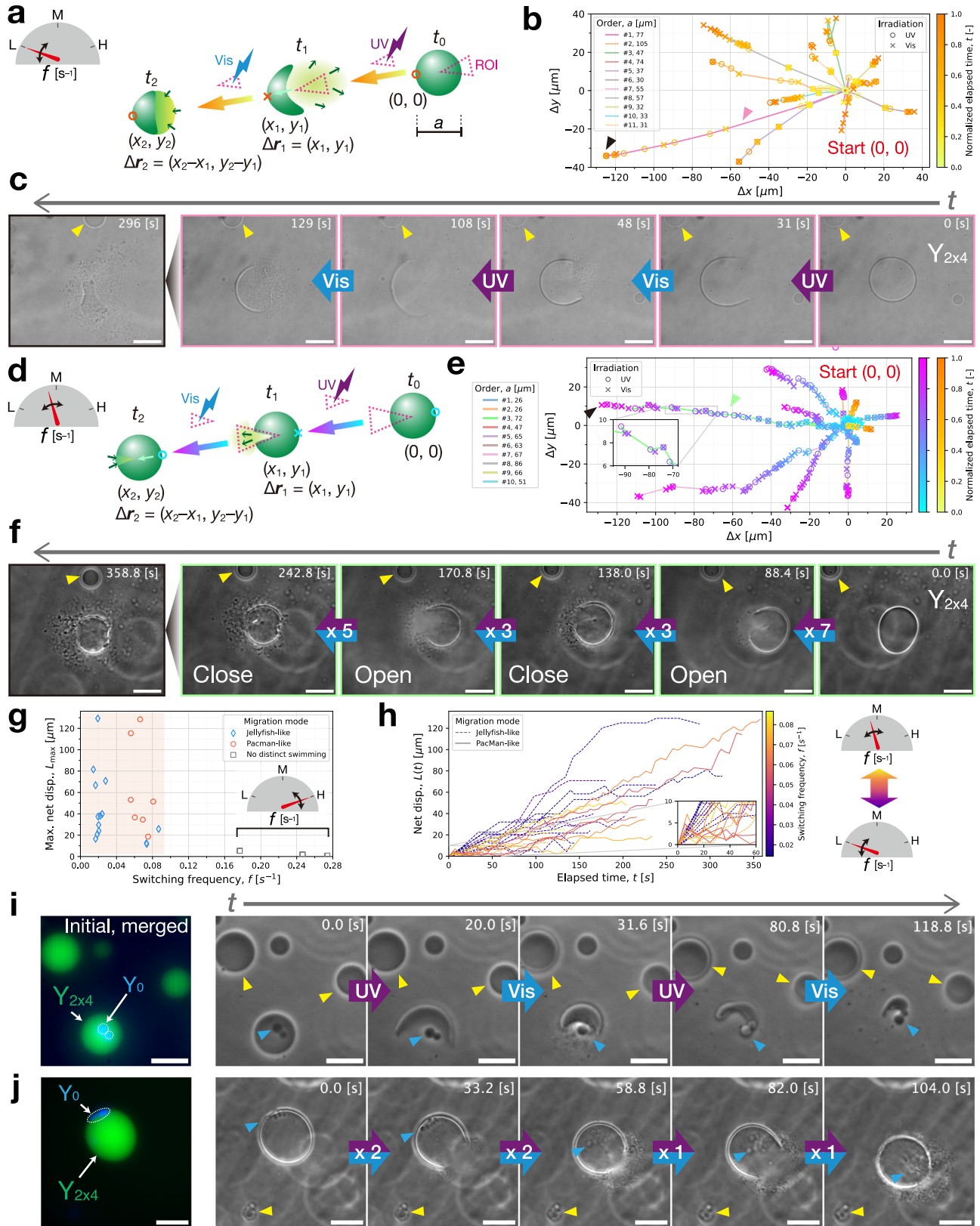

(~15 μm), as the cargo was constantly unwrapped during the alternate opening and closing of the migrating condensate at the aft end. Moreover, the cargo loading deterred push-swimming with a success rate of less than 10%, a significant drop from the cargo-free push-swimming of >50%.

Conversely, a pull-swimming condensate, achieving a final displacement of 70 μm, transported its cargo over a distance exceeding 100 μm (Fig. 7j, Supplementary Movie 10). The cargo loading did not affect the migration efficiency, due to the fore-end opening and closing on the other side of the wrapped cargo. We also noticed significant

**Fig. 7 | Directional motions of DNA liquid condensates (composed of $Y_{2x4}$) with localized UV–Vis photoswitching. a–c**, Push-swimming at low switching frequencies. **a** Illustration of the motion driven by the switched irradiation localized within a region of interest (ROI). The condensate is pushed forward by a UV-spread DNA microflow, which was directed backward while unfurling from the aft end and reversed forward upon switching to Vis. A displacement between two consecutive switching moments, $t_i$ and $t_{i-1}$, is represented as $\Delta\mathbf{r}_i$. Advances of the condensate interface for 11 successive operations, lasting up to completely crumbling, are plotted in (**b**) a two-dimensional (2D) map. A time-evolving displacement $(\Delta x, \Delta y) = \sum_i \Delta\mathbf{r}_i$. For clarity, several of them are rotated around the start point (0, 0). Markers are colored according to a yellow-to-orange color bar that represents an elapsed time normalized by their time durations. **c** Time-sequential images of push-swimming in the first operation (#1) captured at four consecutive UV–Vis alterations. **d–f** Pull-swimming at intermediate switching frequencies. **d** Illustration of pull-swimming. By quickly switching to Vis before the UV-discharged DNA diffuses away, the reversed flow can pull the condensate toward the ROI over a finite

distance. In (**e**) a 2D plot of the advances of the condensate interface observed in 10 successive operations, a cyan-to-purple color bar (left) corresponds to the normalized elapsed time for pull-swimming. **f** Time-sequential images of pull-swimming in the third operation (#3) captured when a condensate opened (UV) and closed (Vis) its fore end, with multiple UV–Vis switching intervals. **g** Scatter plot of maximum net displacements $L_{max}$ for a range of the switching frequencies $f$. A net displacement $L(t)$ is obtained through $L(t) = |\sum_i \Delta\mathbf{r}_i|$. Time evolutions of the data points in the shaded range $L(t)$ are further plotted in (**h**). Cargo transport using **i** push- and **j** pull-swimming of liquid condensates comprising cross-linked motif $Y_{2x4}/L'_0/Y_0$ (Supplementary Table 18). Blue triangles point to $Y_0$ gel-state condensates serving as cargo. (**c, i, j**) Yellow triangles point to immobile nearby DNA liquid condensates as reference objects. Irradiation intensities: (UV, 360–370 nm) 3.3 mW cm$^{-2}$, (Vis, 460–495 nm) 7.0 mW cm$^{-2}$. Temperatures: (**b, e**) 36.2°C; (**i**) 41.5°C; (**j**) 36.8°C. Scale bars: (**c, f, j**) 50 μm; (**i**) 20 μm. Source data are provided as Source Data files.

three-dimensional (3D) rolling of the swimming condensate as it was dragging the embedded cargo. On a closer re-inspection of the cargo-free pull-swimming, a similar rolling behavior was detected in some small irregularities being rolled up on the surface (Supplementary Movie 8).

For a greater understanding of the underpinning mechanisms, we performed numerical simulations to reproduce the observed differences in the lifetime, direction, and stepwise displacement of the swimming behaviors. We built a model that features the mass and energy transport between liquid-state and dissociated-state domains in a DNA condensate subjected to localized photoswitching (Fig. 8a, Supplementary Fig. 12a). In a binary system comprising 'Liquid' and 'Dissociated' states (Fig. 8b), their time-fluctuating masses $m_L(t)$ and $m_D(t)$ were evaluated ($t$: elapsed time from the start). The model simplifies the localized photoirradiation as a pulse-like shot with an infinitesimally short duration. The mass transport is modeled as two opposing alternate transfers as observed in the spread–collect action (Fig. 5g): one in UV-to-Vis intervals is free diffusion of UV-dissociated motifs, and the other in Vis-to-UV intervals is convective flow of Vis-reassociated motifs.

In UV-to-Vis intervals, a fraction $A$ of the constituting motifs in 'Liquid' state, $A \cdot m_L(t)$, generates a spreading flow of dissociated motifs; subsequently, a large fraction $\lambda$ of the spreading flow is diffused away from the cycle reaction. This adds an amount of $(1-\lambda)Am_L(t)$ to $m_D(t)$ per unit time (Fig. 8b). The disassembly ratio $A$ and disengagement ratio $\lambda$ are diffusion-associated parameters $A(t')$ and $\lambda(t')$, which were presumed to grow non-linearly with time $t'$ relative to a characteristic diffusion time $\tau^*$. Be aware that $t'$ is treated as a lap time elapsed in each UV-to-Vis (Vis-to-UV) interval in the model, and hence its time evolution is reset to zero upon photoswitching. In Vis-to-UV intervals, the opposite transfer of a fraction $B$ (constant) of 'Dissociated'-state motifs takes place from $m_D(t)$ to $m_L(t)$ with limited diffusion, adding $(1-\kappa)Bm_D(t)$ to $m_L(t)$ per unit time, where $\kappa$ (constant) means the disengaged ratio in the reversing flow. The alternating mass transfers between the binary states are calculated by the following differential equations:

$$\frac{dm_{L,UV\to Vis}(t)}{dt} = -A(t')m_{L,UV\to Vis}(t), \quad (\text{'Liquid' in UV−to−Vis}) \quad (1)$$

$$\frac{dm_{D,UV\to Vis}(t)}{dt} = (1-\lambda(t'))A(t')m_{L,UV\to Vis}(t), \quad (\text{'Dissociated' in UV−to−Vis}) \quad (2)$$

$$\frac{dm_{L,Vis\to UV}(t)}{dt} = (1-\kappa)Bm_{D,Vis\to UV}(t), \quad (\text{'Liquid' in Vis−to−UV}) \quad (3)$$

$$\frac{dm_{D,Vis\to UV}(t)}{dt} = -Bm_{D,Vis\to UV}(t), \quad (\text{'Dissociated' in Vis−to−UV}) \quad (4)$$

where the subscripts point to the corresponding intervals.

The model can produce an emergent migration of 'Liquid' domain, driven by the mass transport introduced above. With a primary focus on the migration direction, it is restricted to one-dimensional (1D) motion. Two contributions toward the velocity time evolution are featured: one from momentum exchange between 'Liquid'- and 'Disperse'-state domains and the other from thermal dissipation due to viscosity (see Supplementary Texts for the derivation). The 1D velocities of 'Liquid' state $u(t)$ in UV-to-Vis and Vis-to-UV intervals are dictated respectively by

$$\frac{du_{UV\to Vis}(t)}{dt} = \frac{du^M_{UV\to Vis}(t)}{dt} + \frac{du^T_{UV\to Vis}(t)}{dt}, \quad (\text{'Liquid' in UV−to−Vis}) \quad (5)$$

$$\frac{du_{Vis\to UV}(t)}{dt} = \frac{du^M_{Vis\to UV}(t)}{dt} + \frac{du^T_{Vis\to UV}(t)}{dt}, \quad (\text{'Liquid' in Vis−to−UV}) \quad (6)$$

where the superscripts denote the momentum exchange ($M$) and thermal dissipation ($T$). Briefly, $du^M_{UV\to Vis}(t)/dt$ in the righthand side (RH) of Eq. (5) refers to propulsive effect due to freely diffusive UV-discharged microflow in UV-to-Vis intervals, whereas $du^M_{Vis\to UV}(t)/dt$ in the RH of Eq. (6) relates to the momentum transfer from 'Dissociated' to 'Liquid' domains within Vis-to-UV intervals. Importantly, this momentum from 'Dissociated' domain consists of two contributions: one is the Vis-generated reversing flow directed toward the ROI in the current Vis-to-UV interval; the other originates from UV-discharged microflow that is directed away from 'Liquid' domain in the previous UV-to-Vis interval. The dissipation terms, $du^T_{UV\to Vis}(t)/dt$ and $du^T_{Vis\to UV}(t)/dt$, have the same form in common, where the viscous force counteracts the migration in proportion to the current velocity. Simulations were performed at low, intermediate, and high switching frequencies $f$.

The simulated mass of 'Liquid' domain $m_L(t)$ showed a longer lifetime with increasing $f$ (Fig. 8c). Further, the low-$f$ migration exhibited intense fluctuations in the velocity $u(t)$, which were predominantly in the positive range. In contrast, the intermediate-$f$ migration displayed negatively signed fluctuations with lower peak heights. The overall displacement $|x|$ in the elapsed time simulated was larger at the intermediate $f$ than at the low $f$. At the highest $f$, we obtained a minimum displacement over the elapsed time simulated. These numerical results in the lifetime, migration direction, and interval displacement were consistent with the observed swimming behaviors.

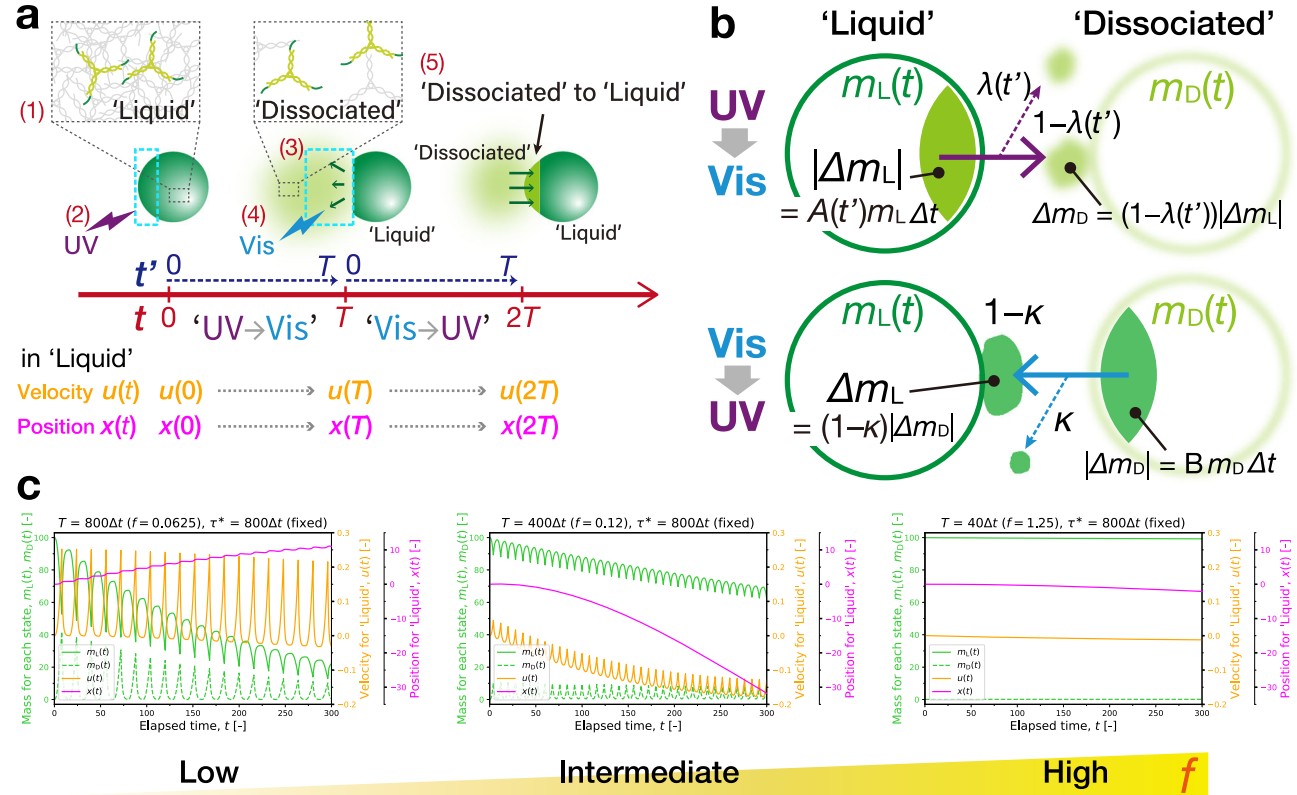

**Fig. 8 | Numerical simulations of the directional motion. a** Representative schematic of the model used. Photoreversible 'Liquid' and 'Dissociated' states in a DNA condensate undergo events (1) – (5) over two intervals, UV-to-Vis and Vis-to-UV, each with an interval length $T$. Besides an elapsed time $t$ from the start, a lap time $t'$ periodically runs from 0 to $T$ within each interval. $u(t)$ and $x(t)$ are the 1D velocity and position, respectively, of 'Liquid' domain. It is driven by **b** photoswitched mass transport between the binary states during each interval. The alternate opposite transfers over a time step $\Delta t$ are illustrated. The masses of the binary states are respectively denoted as $m_L(t)$ and $m_D(t)$. $A(t')$ and B (constant) are disassembly ratios, which refer to a fraction of the constituting motifs being transferred from one state to the other per unit time. Halfway through the transfers, the state-switching motifs diffuses away from the cycle reactions with disengagement ratios: $\lambda(t')$ in UV-to-Vis and $\kappa$ (constant) in Vis-to-UV intervals. **c** Simulated time evolutions of the masses of the binary states, $m_L(t)$ and $m_D(t)$, and the velocity $u(t)$ and position $x(t)$ of 'Liquid' domain. Low, intermediate, and high switching frequencies $f$ are highlighted. Simulations started with $m_L(t) = 100$, $m_D(t) = 0$, and $\Delta t = 0.01$. $t^*$ refers to a characteristic diffusion time of $800\Delta t$.

The analogy between the numerical and experimental results in these aspects suggests the following: First, the longer-lived migrations at higher $f$ (negatively signed) are linked to a smaller amount of diffusive loss of the mass. The positively signed migration is driven by propulsive force due to UV-discharge microflow, wherein the mass of the condensate continues to be consumed with a non-linearly growing disassembly ratio $A(t')$. Further, the diffusion-associated disengagement of the motifs from the cyclic reactions in UV-to-Vis intervals, $\lambda(t')$, grows non-linearly with time, in contrast to $\kappa$, the counterpart of Vis-to-UV intervals, assumed as constant. Hence, within tighter intervals (higher $f$), the mass dissipation becomes disproportionly less prominent, resulting in longer-lived migration.

Another important indication is the contribution of momentum exchange between 'Liquid' and 'Dissociated' states to the sign inversion in the migration direction. To show this contribution, a numerical comparison was made between the cases with and without the momentum exchange in Vis-to-UV intervals. We observed that the lack of momentum exchange eliminated the $f$ dependence of the migration direction, where positively signed push-swimming was dominant at varied switching frequencies (Supplementary Fig. 13). This suggests that the momentum transfer from 'Dissociated' to 'Liquid' domains was essential to negatively signed pull-swimming. In UV-to-Vis intervals, the condensate consumes its mass with a non-linearly growing ratio $A(t')$, which transforms into a repulsive force. This results in a non-linear growth of positively signed push-swimming with an elongated interval (lower $f$). In a tighter interval (higher $f$), among the momentum transferred from 'Dissociated' to 'Liquid' domains in a Vis-to-UV interval, the negatively signed momentum that has originated in the previous UV-to-Vis interval is more conserved. This means that the microflow is more likely to pull 'Liquid' domain toward the ROI in the tighter interval. Thus, the switching frequency $f$ is key to adjusting the weight of the energy-dissipating and energy-exchanging contributions in the mass and energy transport.

Assuming that the switching frequency determines between energy-dissipating push-swimming or energy-exchanging pull-swimming, a threshold switching frequency for pull-swimming $f_c$ can be roughly estimated. For instance, consider a condensate with a size $a$ of 20 μm, with the resulting microflow possessing effective diffusion coefficients $D_S$ for the UV-induced spreading (Fig. 5h(i)) and $D_R$ for the Vis-reversed flow (Fig. 5h(ii)). From the experimental measurements of DNA microflow mobilities $D^*$ using the 2-nt longer SE (Fig. 6f), we may reasonably adopt $D_S$ of 20 μm s$^{-1}$ (from 'UV' panel) and $D_R$ of 10 μm s$^{-1}$ (from '+Vis' panel). Now, UV-discharged microflow spreads a distance of the condensate size, and upon switching from UV to Vis, the reversed flow travels the same distance backward. Referring to the well-known relation of mean square displacement $\langle r^2(\tau) \rangle$ ($\tau$: a time interval) with diffusion coefficient $D$ via $\langle r^2(\tau) \rangle \sim 2dD\tau$, where dimension $d = 2$ (2D), we can calculate the characteristic cycle time $\tau_c$. It is defined as the sum of a characteristic time for UV-induced diffusion $\tau_{UV}$ and

that for Vis-enabled reversed flow $\tau_{Vis}$ over a distance of the condensate diameter. This gives $\tau_c = \tau_{UV} + \tau_{Vis} = a^2/4D_S + a^2/4D_R = 15$ s, corresponding to $f_c \sim 0.07\,s^{-1}$. This approximate estimate falls within the pull-swimming regime in Fig. 7g. This suggests that when $f < f_c$ (a long interval), the photoswitched liquid condensate is under the dissipation-dominant control; when $f > f_c$ (a tight interval), it is under the energy-exchanging-dominant control.

Pull-swimming behavior at intermediate $f$ still requires further elucidation. Apparently, from Fig. 7f, the microflow generated by a fraction of the total mass kept pulling the denser, unaffected domain of the liquid condensate. This raises the question of why the Vis-reversed flow could pull, rather than be dragged toward, the denser liquid body. The answer must also consistently address the rolling behavior observed in the cargo transport by pull-swimming (Fig. 7j).

## Discussion

We demonstrated remote-controlled microflow of DNA condensates capable of multi-modal mechanical actions and directional motions. We endowed the DNA condensates with photoresponsive capabilities by introducing Azo residues in the SEs as the on/off signal transmission and energy injection mechanisms. The DNA condensates exhibited a well-controlled macroscopic state transition between the gel, liquid, and dissociated states and thus a fluidity change in a wavelength-dependent manner. The PAGE and FRAP experiments provided solid support for the relationship between the irradiation-determined SEs binding state and the resulting motif mobility. Photoswitching SEs binding state resulted in non-equilibrium microflow of DNA, which was drastically mode-switchable only via several-base relocation of Azo residues in the SE. The multi-modal DNA microflow was able to perform mechanical actions upon the sequence-specifically specified DNA by coupling photoresponsive and nonresponsive motifs by a branched cross-linker. The mechanical actions observed depended on Azo insertion sites in the SE, which were classified into (1) radial spreading (in UV), (2) elastocapillarity-driven collapsing (in UV), and (3) spread–collect motion (UV–Vis switching). To elucidate the thermodynamic origin of the multi-modality, the enthalpy differences in the Azo-modified SEs binding, $\Delta H_{trans}$ and $\Delta H_{cis}$, were estimated by measuring $T_D$ in the resulting condensates of various Azo-free motifs. The photoinduced shift of $|\Delta(\Delta H)_{trans \to cis}|$ in the phase map spanned by $|\Delta H| - T$ space, which was determined by the Azo insertion site, was suggested as a determinant of the multi-modality of DNA microflow. Localized UV–Vis switching within a photoresponsive DNA condensate created two modes of transient yet pronounced swimming behaviors: push- and pull-swimming. Depending on switching frequencies, these modes showed contrastingly different morphological dynamics, lifetimes, migration directions, and interval displacements. For a deeper understanding of the directional motions, we modeled the mass and energy transport between the liquid- and dissociated-state domains of a liquid condensate subjected to localized photoswitching. The numerical results revealed the switching frequency as a regulatory parameter in determining between energy-dissipating push-swimming and energy-exchanging pull-swimming.

Overall, UV or Vis irradiation that provides a signal (for switching on/off the motif binding) as well as energy dynamically changed the enthalpy difference $\Delta H$ between the bound–unbound states in the SEs (Supplementary Fig. 10). This molecular-level thermodynamic shift translated into the multi-modal mechanical motions of the cross-linked motif condensates (Fig. 5) and the directional push- or pull-swimming behaviors of locally photoswitched liquid condensates (Fig. 7).

Our DNA-based microfluid regulation can be widely applied within confined biological/biomimetic environments that are inaccessible with conventional top-down microfluidics methods. Notably, our method only requires sequence programming and simple fine-tuning of the photoreactive agents in the SEs. Therefore, our method would provide a facile mechanistic tool for microscale fluid manipulation. Our method does not rely on any complex use of multiple chemical compounds or time-consuming manufacturing processes compared to the Janus particles extensively studied so far[76,78]. This energy-transducing DNA condensate system would see a rich variety of applications as a programmable microfluidic system.

## Methods

### DNA motif construction

**Sequences**. The photoresponsive Y-motifs and cross-linked DNA systems were designed using the oligonucleotides listed in Supplementary Tables 1–18. The basic sequence designs were referenced from a previous study[24]. The DNA motifs were equipped with photoresponsive capabilities by introducing Azo into the SEs.

Azo-modified oligonucleotides were purchased from Tsukuba Oligo Service (Ibaraki, Japan), and the other non-Azo-modified strands from Eurofins Genomics (Tokyo, Japan). The former sequences were purified with high-performance liquid chromatography (HPLC), and the latter sequences were purified with an oligonucleotide purification cartridge (OPC). The oligonucleotides were dissolved in ultrapure water (Milli-Q, 18.2 MΩ cm, Direct-Q UV 3 with Pump, Merck KGaA, Darmstadt, Germany) at a concentration of 100 μM upon arrival at the laboratory and stored in a − 30°C freezer until use.

**Material composition**. In Y-motifs (Supplementary Tables 1–4), an equimolar mixture of three ssDNAs to jointly form Y motifs was dissolved in a test tube at a final concentration of 5.0 μM in a buffer containing 350 mM NaCl ( > 99.5% purity, FUJIFILM Wako Pure Chemical Corp.) and 20 mM Tris-HCl pH 8.0 (UltraPure, Thermo Fisher Scientific, MA, US). For confocal microscopy, a fluorescent dye (SYBR Gold Nucleic Acid Gel Stain, Thermo Fisher Scientific) was added to the buffer at a 1× concentration.

In cross-linked DNA systems (Supplementary Table 10), a mixture of Azo-containing ssDNAs (15 μM $Y_i$-1, -3 and 13.5 μM $Y_i$-2) with 1.5 μM $Y_i$-2_FAM and non-Azo-containing ssDNAs (5 μM $Y_0$-1, -3 and 4.5 μM $Y_0$-2) with 0.4 μM $Y_0$-2_Alexa405 and 0.1 μM $Y_0$-2_Cy3 to jointly form a sequence-specifically photoresponsive DNA motif was dissolved in the same buffer similar to that of Y-motifs. A 10% fraction of $Y_i$-2 and $Y_0$-2 were the dye-labeled strands. The concentration ratio of $Y_0$-2_Alexa405 to $Y_0$-2_Cy3 was set at 4:1 to improve the detectability of Alexa405, whose effective excitation wavelength deviates from that of UV irradiation applied. Note that $i = $ 1x7, 3x5, 2x1x5.

In Y-motif for the directional motion (Supplementary Table 5), a mixture of Azo-containing ssDNAs (15 μM $Y_{2x4}$-1, -3 and 14.4 μM $Y_{2x4}$-2) and 0.6 μM $Y_{2x4}$-2_FAM was dissolved in the buffer. To demonstrate cargo transport using the directional motion (Supplementary Table 18), a mixture of Azo-containing ssDNAs (5 μM $Y_{2x4}$-1, -3 and 4.5 μM $Y_{2x4}$-2) with 0.5 μM $Y_{2x4}$-2_FAM was dissolved in the buffer. As cargo, non-Azo-containing ssDNAs (3 μM $Y_0$-1, -3 and 2.7 μM $Y_0$-2) with 0.3 μM $Y_0$-2_ Cy3 were dissolved in the buffer.

**Annealing protocol**. Annealing schedules were programmed and executed on a thermal cycler (Mastercycler Nexus X2, Eppendorf, Hamburg, Germany). For Y-motifs (Supplementary Tables 1–4), DNA solutions were heated from RT up to 85°C with a ramp of +1.0°C/min, held for 3.0 min, and cooled down to 25.0°C with −1.0°C/min. This annealing procedure is referred to as "short annealing". For cross-linked DNA systems (Supplementary Table 10), DNA solutions were heated from RT up to 85°C with a ramp of +1.0°C/min, held for 3.0 min, cooled down to 64.0°C with −1.0°C/min, held for 30 min, cooled down to 56°C with −0.25°C/min, and quickly cooled down to 25.0°C with −2.0°C/min. This annealing process is called "long annealing". For the directional motion of Y-motif condensates, a mixture of $Y_{2x4}$-1, -2, -3

was subjected to the long annealing. In the cargo transport, $Y_{2x4}$ and the cargo solutions underwent the long and short annealing procedures, respectively. The annealed cargo solution was diluted 4-fold with the buffer. Then, the $Y_{2x4}$ and the cargo solutions were mixed at a ratio of 3:1, and incubated at RT for 1 h.

## Microscopy observation
**Observation chambers.** (Fig. 2, Fig. 3, Supplementary Fig. 10c, and Supplementary Movies 1, 2) A 30-mm × 40-mm No.1 glass plate (thickness 0.13–0.17 mm, Matsunami Glass Ind., Ltd., Kishiwada, Japan) was treated with bovine serum albumin (BSA, FUJIFILM Wako Pure Chemical Corp.) coating. First, the glass was soaked and shaken for > 15 min in a BSA-containing solution (5 w/v% BSA and 20 mM Tris-HCl). As a spacer, a 20-mm × 20-mm square piece was cut from a silicone rubber sheet (1 mm × 300 mm × 300 mm, Tigers Polymer Co., Ltd., Osaka, Japan), and a 5-mm-diameter hole was pierced into it. Next, after the spacer was affixed to the glass plate, a sample was applied within the pierced hole, which was then filled with mineral oil (Nacalai Tesque, Inc., Kyoto, Japan) to avoid dehydrating the sample.

(Figures 5, 7 and Supplementary Movies 3–10) Samples were confined tightly between glass plates to minimize unwanted flow disturbance. They were sandwiched between a BSA-treated 30-mm × 40-mm No.1 glass plate and an 18-mm × 18-mm No.1 cover glass (Matsunami) spaced by double-sided tape. Subsequently, the resulting confined space was filled with mineral oil via capillary sucking. Finally, the opening slits between the plates were sealed using nail polish.

**Temperature control.** (Fig. 2e and Supplementary Movies 1, 2) During the confocal microscopy observation, the sample-harboring pierced-hole chamber was placed on a Peltier heating stage (10021-PE120, Linkam Scientific Instruments Ltd., Surrey, UK). In the other observations, during the fluorescence microscopy observation, the sample-sandwiching chamber was placed on a thermoplate (TPi-110RX, Tokai Hit Co., Ltd., Fujinomiya, Japan).

**Imaging.** (Fig. 2e and Supplementary Movies 1, 2) The samples were visualized using a confocal laser scanning microscope (FV1000, Olympus, Tokyo, Japan) using a 50× long-working-distance objective (LMPLFLN, Olympus) was employed. In the other images, we used a fluorescence microscope (IX71, Olympus) for fluorescence and phase-contrast (PC) imaging with a 60× objective lens (LUCPlanFL N, Olympus). IX71 was also used for $T_D$ evaluation of the photoresponsive condensate (Fig. 3). Snapshots and time-sequential images were captured with a Zyla 5.5 (Andor Technology, Belfast, UK) mounted on the IX71.

## Native PAGE
**Single-SE Y-motifs.** In the control experiments, Azo-free single-SE DNA motif was used (Supplementary Table 6); photoresponsive single-SE DNA motifs with $SE_i$ ($i = 1x7$, 3x5, 2x1x5) were considered for the irradiation effects on SEs' photoswitchability (Supplementary Tables 7–9). Triple-SE motifs as shown above would form network-structured condensates, which prevents well-distinguishable binary states. Thus, we expected UV and Vis irradiation on the single-SE motifs to enhance the lower and upper bands, respectively. The annealing process and buffer conditions for the single-SE motifs were identical to those for triple-SE Y-motifs.

**Gel electrophoresis.** A mixture of 2.5-mL 40 w/v%-Acrylamide/Bis Partitioned Solution (29:1, Nacalai Tesque) and 2.0-mL 5× Tris-borate EDTA Buffer (TBE, Nippon Gene, Toyama, Japan) was topped up to 10 mL with Milli-Q water in a test tube to prepare an 8-cm × 8-cm gel plate with 10% concentration. Subsequently, the tube was gently inverted several times after adding 150 μL of 10 w/v% APS (ammonium persulfate, FUJIFILM Wako Pure Chemical Corp.). After adding 3.0-μL N,N,N',N'-tetramethylethylenediamine (TEMED, FUJIFILM Wako Pure Chemical Corp.), the tube was inverted similarly. The solution was incubated for 60 min in an assembled gel electrophoresis cassette for polymerization.

For each lane in Control, $SE_{1x7}$, $SE_{3x5}$, and $SE_{2x1x5}$ (Supplementary Fig. 4), an equimolar 1.0-μL mixture of the photoirradiated sample and 2× loading buffer was loaded within a designated well in a 4 °C room. This low-temperature condition was preferred because PAGE results performed at room temperature (RT) showed less distinct band images. A 100-bp ladder (Quick-load, New England Biolabs, Ipswich, MA, US) was loaded at 0.8 μL in the rightmost lane. The 2× loading buffer contained 1.0 w/v% bromophenol blue (BPB, FUJIFILM Wako Pure Chemical Corp.), 10 v/v% glycerol (FUJIFILM Wako Pure Chemical Corp.), and 50-mM ethylenediaminetetraacetic acid (EDTA, FUJIFILM Wako Pure Chemical Corp.). Next, gel electrophoresis was conducted in the 4 °C room with a PowerPac Basic Power Supply (Bio-Rad Laboratories, CA, US) for 80 min at a constant voltage of 40 V. Finally, for fluorescence staining, the gel plate was soaked and gently shaken at RT for 5 min in 100-mL 1× TBE containing 0.01 v/v% SYBR Gold. The gel plate was imaged with a Gel Doc EZ (Bio-Rad Laboratories).

**Quantification of the bands.** The images acquired of the gel plates were subjected to quantitative analysis of the staining intensity using Fiji. Within a selection rectangle covering a stained band, the "Integrated Density" ("IntDen") command was executed to consider the concentration of the corresponding motif state (monomer or dimer).

## UV/Vis irradiation
(Figures 2, 3, Supplementary Fig. 4, and Supplementary Movies 1, 2) A xenon lamp (300 W, MAX-303, Asahi Spectra, Tokyo, Japan) was used as a light source for photoswitching DNA condensates. Irradiation intensities were measured using a power meter (3664, Hioki E. E. Corp., Ueda, Japan) connected with a lightweight sensor (9742, Hioki E. E. Corp.). In the gel-to-liquid phase change during PC microscopy imaging (Fig. 2a), Azo/non-Azo DNA gel particles were irradiated by UV (365 nm) of 1.1 mW cm$^{-2}$ for the initial 60 min and 0.60 mW cm$^{-2}$ for further irradiation. In the dissociated-to-liquid phase change (Fig. 2b), DNA condensates were initially dissociated by UV (8.9 mW cm$^{-2}$) until disappearance, followed by Vis (440 nm, 10.4 mW cm$^{-2}$) until recondensation. In the FRAP experiments (Fig. 2e), the FAM-labeled DNA gel was irradiated with UV (8.9 mW cm$^{-2}$) and Vis (10.4 mW cm$^{-2}$). In the confocal imaging (Supplementary Movies 1, 2), the irradiation intensities used were 0.34 mW cm$^{-2}$ for UV and 7.9 mW cm$^{-2}$ for Vis. In the PAGE experiments (Supplementary Fig. 4), solutions of single-SE motif were encapsulated in glass capillaries (ID: 1.12 mm, OD: 2 mm; World Precision Instrument, FL, US), because of their high UV transmittance and ease of handling a minute amount of liquid (Supplementary Fig. 1). In the irradiation procedures, the glass capillaries were placed on a black metal plate to avoid the visible light reflection of UV light from the surface. The irradiation intensities were 2.0 mW cm$^{-2}$ (UV) and 3.9 mW cm$^{-2}$ (Vis). In the $T_D$ measurements (Fig. 3), DNA condensates were irradiated with Vis (4.4 mW cm$^{-2}$, *trans*) for 5 min and UV (0.34 mW cm$^{-2}$, *cis*) for 10 min.

(Figures 5–7, Supplementary Fig. 6, and Supplementary Movies 3–10) We observed the photogenerated migrations of the cross-linked DNA systems using the IX71. We set the excitation filter to 360–370 nm (UV, 3.3 mW cm$^{-2}$) for the *trans*-to-*cis* isomerization of Azo in the photoresponsive $Y_i$ and the excitation of Alexa405 introduced in the non-photoresponsive $Y_0$. For the excitation of FAM in $Y_i$, the excitation filter was set to 460–495 nm (7.0 mW cm$^{-2}$). Additionally, the filter was set to 520–550 nm (Vis, 4.8 mW cm$^{-2}$) for the *cis*-to-

*trans* isomerization of Azo in $Y_i$ and the excitation of Cy3 introduced in $Y_0$.

In Fig. 7, the localized irradiation was achieved using Mosaic3 (Andor Technology), a programmable DMD, attached to the IX71. UV–Vis irradiation was localized within a triangular or circular ROI approximately half the size of a targeted condensate, which was specified using iQ, driver software provided by Andor. UV–Vis switching was performed manually by alternating between mirror cubes in the filter wheel. Below the moderate switching frequencies $f < 0.16\,[\text{s}^{-1}]$, the time duration of a single irradiation was fixed at 2.2 s, while constant irradiation was applied in the higher-frequency regime.

### Measurement of the dissolving temperature $T_D$

In Fig. 3, we irradiated the samples with Vis light in the initial step and observed $T_D$ for *cis/trans* Azo isomers while exposing them to UV/Vis, respectively. Different combinations of microscopes and heating stages were used as follows: (1) the confocal microscope (FV1000) and the Peltier heating stage for the samples whose $T_D$ were apparently greater than 60°C; and (2) the fluorescence microscope (IX71) and the thermoplate (Tokai Hit). The thermoplate was calibrated using its auxiliary temperature sensor.

We observed Control, SE$_{1x7}$ (*trans*), and SE$_{3x5}$ (*trans*) using the combination of (1) the FV1000 and the Peltier heating stage. After heating up to a given temperature and a subsequent 5 min incubation, we examined whether DNA condensates were dissociated in the buffer while scanning the sample with the 488-nm laser. When the DNA condensates were invisible, the temperature was determined to be $T_D$. Otherwise, the applied temperature was increased by approximately 1°C, and the same procedure was repeated until complete dissolution was detected. For $T_D < 60$°C, we observed SE$_{1x7}$ (*cis*), SE$_{3x5}$ (*cis*), and SE$_{2x1x5}$ (*cis/trans*) using the combination of (2) the IX71 and the thermoplate. The samples were examined using PC microscopy with the background illuminated as weakly as possible to minimize the Vis effect from the bright field illumination. The UV (360–370 nm) and Vis (520–550 nm) beams emitted from the objective were employed to maintain the *cis/trans* states for the isomerization reaction. In the case of Y$_{2x1x5}$ (*cis*) with $T_D$ of approximately 3°C, some ice blocks were placed near the sample to maintain the surrounding temperature below 10°C. In Supplementary Fig. 10c, the same heating stages as in Fig. 3 were used for different temperature ranges.

### Measurement of the MSD and signed diffusion coefficient $D^*$

**Trajectory detection.** The time-series data of the photogenerated migrations recorded by the IX71 (frame interval of 200 ms and exposure of 100 ms) were subjected to particle tracking analysis using TrackMate[75], an open-source plugin in Fiji. In the parameter settings, we selected the LoG detector for the particle detection and the Simple LAP tracker for the particle linker (Supplementary Fig. 8). Therefore, to maximize the duration of detected paths, the track filter was adjusted to select the tracks spanning the entire recording time.

**MSD.** We calculated the MSDs of individual trajectories in each experiment, including tens to hundreds of the detected trajectories, from the 2D trajectories acquired above. Additionally, to reasonably reduce the computational cost for large calculations, we referred to a useful algorithm that yields MSDs through Fast Fourier Transform[79]. Its underlying idea is to use the Wiener–Khinchin theorem, which relates the power spectrum of a given signal with its auto-correlated function via the Fourier transform. To focus on the significant MSD curves, the obtained MSD curves as a function of time interval $\tau$ were reduced to the first 20% of the entire time data points. Up to 20 MSD plots with the steepest curves were sorted out in each plot for the representative demonstration of the MSDs (Fig. 6c, d).

$D^*$ **plot.** Linear curve fitting was applied to the data points in the largest 20% $\tau$ of each MSD plot to determine the unsigned diffusion coefficient $D'$ from the fitted slope of $4D'$ (Supplementary Eq. 2). Subsequently, up to 20 MSD curves with the largest $D'$ values were sorted out in each case to obtain signed $D^*$ in Fig. 6f. Lastly, the direction of the generated migrations was determined qualitatively. Four experiments were included ($n = 4$) at each temperature examined.

### Directional motion of DNA condensates by localized UV–Vis switching

**2D tracking.** The directional motion was quantified by tracking the interface of a swimming condensate at each UV-to-Vis (or Vis-to-UV) switching captured in the time-sequential images. When the field of view was repositioned between two consecutive frames, an immobilized unirradiated object in the overlapping pixels was selected as the reference. Its pixel values were set to zero as a marker to facilitate the Fiji 2D auto-stitching.

**Numerical simulation.** A Python script was coded to iteratively solve the time-dependent fluctuation of the liquid and dissociated states in a DNA condensate under localized photoswitching and the resulting migration velocity (see 'Code availability'). The iterative calculations were performed with the Euler method. Further details of the model, governing equations, and input parameters are provided in the Supplementary Texts.

## Data availability

The datasets supporting the key findings in this work are available within the main text, Supplementary Information, and Source Data files. Source Data files are provided with this paper, deposited in a public repository (https://figshare.com/projects/Remote-controlled_mechanical_and_directional_motions_of_photoswitchable_DNA_condensates_H_Udono_et_al_Nat_Comm_2025_/244145), together with the relevant Python scripts used to generate the plots. All data relevant to Source Data are available from the corresponding author upon request.

## Code availability

The Python codes used for the numerical simulations (Fig. 8 and Supplementary Fig. 12) are available at GitHub (https://github.com/takinouelab/UdonoTakinoue2024, https://doi.org/10.5281/zenodo.15174276).

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

## Acknowledgements

We thank Dr. Marcos K. Masukawa and Dr. Yu Fujio (Institute of Science Tokyo, "Science Tokyo"), Prof. Yusuke Sato (Kyushu Institute of Technology), Prof. Hirohide Saito (The University of Tokyo), and Prof. Yoshihiro Shimizu (RIKEN) for instructive discussion and suggestions. We also appreciate useful experimental tip instructions by Ms. Yukiko Okuda, Dr. Jing Gong, and Dr. Tomoya Maruyama (Science Tokyo). This work was supported by MEXT/JSPS KAKENHI (No. JP20H05701 to S.M.N. and M.T., Nos. JP20H00619, JP20H05935, and JP24H00070 to M.T., and No. JP22K14528 to H.U.), JSPS Grant-in-Aid for JSPS fellows (Nos. JP19J112959 and JP22J00940 to H.U.), Human Frontier Science Program (HFSP; RGP0016/2022–102 to M.T.), and JST Adopting Sustainable Partnerships for Innovative Research Ecosystem (ASPIRE) (No. JPMJAP24B4 to M.T.).

## Author contributions

H.U., S.M.N., and M.T. contributed equally to the conceptualization and research design. H.U. conducted experiments, simulations, and data analysis, and wrote the first draft of the manuscript. H.U., S.M.N., and M.T. revised the manuscript.

## Competing interests

The authors declare no competing interests.
