## [Peer Review file · Nature Communications]

Remote-controlled mechanical and directional motions of photoswitchable DNA condensates

Corresponding Author: Professor Masahiro Takinoue

Version 0:

Reviewer comments:

Reviewer #1

(Remarks to the Author)

The manuscript describes a molecular system in which the properties of DNA-based materials are photo-switched using the already widely and frequently reported azobene-modified sticky-end strategy. Particularly, DNA materials are photo-switched between gel and liquid states as previously published including in references cited in the manuscript. The uses of the terms “non-equilibrium microflow”, “microfluidic mechanical action”, “cargo and transporter”, “gather”, and “fetch” seem like spin or hype or marketing language rather than some new, important science or engineering. Simply put, thermal heating plus steric (trans Azo) melting of sticky-end adhesion in DNA gels results in enhanced diffusion in liquid states, much of the rest of the rhetoric seems like window-dressing. While it is somewhat difficult to see the morphology changes of the blue/green “fluid” in the still images (for example Fig 4d, Fig 5), the enhanced diffusion during gel melting is much more obvious in the Supplemental videos.

Authors should define “reentrancy” within this context. This reviewer does not appreciate the phrase “transient non-equilibrium state”, because it appears to refer to external perturbation of the system (e.g., irradiation) followed by re-establishment of equilibrium. The unperturbed systems are always tending toward thermodynamic equilibrium.

Melting of the gel proceeds well (Fig 1c), but reformation of the gel is very inefficient (Fig 1e). This appears to be due to dilution of the Y-motifs (or individual oligonucleotides) in surrounding buffer following melting of the gel. Individual sticky-end adhesions are shown to function as planned (electrophoresis results Fig 2b), but large scale gel formation may require a thermal anneal. This point is glossed over in the Results and Discussion with the statement “prescribed annealing process caused a set of three single-stranded DNAs (ssDNAs) to self-assemble into a Y-motif”. The gel state forms efficiently via thermal anneal, but not so well during VIS irradiation.

The authors use the term “DNA fluid” to indicate both the gel and liquid states. This seems strange. Fluids are typically liquids (or gases) (i.e., fluids flow). Why are the gel states also referred to as fluids? The term “fluid” is never defined in a concise and satisfactory way. Phrases like “two immiscible DNA fluids” used when referring to two gel particles seems odd. Yes, diffusion is slow (but non-zero) in gel states and faster (but finite) in liquid states, but this terminology labels the behavior as if it were absolute, black and white.

Comparing “Before UV” images in Fig 5c,d with those in Fig 5e,f it is not at all clear why the starting states are so different (i.e., condensed sphere in c,d and diffuse toroid in e,f). Explain further.

Correct typographical errors in Figures: Trasnporter (Fig 5), complimentary (Fig 4) (should be complementary), liqid (Fig 4)

Reviewer #2

(Remarks to the Author)

The paper describes interesting results demonstrating the fabrication of photo-responsive DNA fluids that undergo light-stimulated phase transitions. The DNA liquids consist of azobenzene-modified toehold Y-shaped DNA structure that undergo gel-to-liquid and dispersed states upon photoisomerization of the trans-azobenzene units to the cis-state. The authors discuss the effects of the number of photo-isomerizable units, their position (in the toehold domains) and their ordering (in the toehold domain) on the phase transition events. The photo-induced phase transition events are, then,

applied to design programmed micro-flow processes. The authors prepare two separate DNA liquids, where one is photo-responsive and the other is not photo-responsive. Upon mixing the DNA liquids and the addition of a six-arm DNA “star,” crosslinking of the DNA liquids proceeds to yield hybrid structures, exhibiting different flow modes controlled by the structure of the composite DNA fluids and the temperature. The non-light-responsive DNA liquid acts as “cargo,” whilst the photo-responsive liquid acts as the transporter. The authors demonstrate a light/temperature dependence of the micro-flow mobility controlled by the structure of the transporter liquid. The experimental results are followed by a physical model relating the enthalpy of the local DNA sticky end changes to the microscopic phase transition phenomena.

The paper represents a high-quality scientific report that justifies publication in Nature Communications. The following comments should be addressed, however, prior to acceptance of the paper.

1. The entire study of light-induced crosslinking of the Y-shaped structures includes the coupled effect of light/temperature on the crosslinking. The authors support the crosslinking events by a molecular level system presented in Figure 2 that includes only light as the crosslinking motive. The temperature used in this experiment is not given, and then the relevance of this model system to the liquid DNA assembly is questionable. The authors should discuss and explain this point. Also, the results presented in Figure 2(C) show very small differences. Error bars are missing to justify the conclusions.
2. The phase transition relies on temporal confocal fluorescence imaging and lacks quantitative evaluation. Supporting the conclusion by quantitative temporal viscosity changes, turbidity and light scattering experiment should be provided.
3. The illumination time-intervals of the model system shown in Figure 2 proceed on a time-scale of minutes. On the other hand, the dynamic mobility experiment shown in Figure 5 and Figure 6 includes irradiation time intervals of seconds (and relatively high temperatures). In general, a counter effect of thermal cis-to-trans isomerization of the azobenzene takes place, and the effect of this process on the phase transition is ignored. A solid discussion on the effects of the thermal cis-to-trans transition of the azobenzene unit on the phase transition processes should be provided.
4. Minor comments
 - Figure 5(e) the gathered Y-Motif should be colored blue.
 - Figure 6(a,b) scale bars should be added.

Reviewer #3

(Remarks to the Author)

The manuscript entitled “Photoinduced gel–liquid transition of DNA fluid generates remote-controlled nonequilibrium microflow” by Udonon et al describes the mechano-thermal behavior of mixtures of hydrogel-forming DNA nanostars, a fraction of which photo-activable. The authors report that upon cycling in temperature, unresponsive gel domains embedded in a photoresponsive matrix can either dissolve or further concentrate depending on molecular details.

While the topic of light control of the aggregation state of a DNA hydrogel can be in principle interesting, the manuscript is, in my opinion, not worth publishing. The manuscript is bombastic, watering down the potentially interesting scientific findings within pages of pompous claims, such as “an exceptional technique for versatile, simple microflow manipulations with mode switchability” (there are tens of these).

The authors show that upon irradiation the DNA gel transforms into either a liquid (see question below), or in a “gas”. This is interesting but it is not a “energy- transducing systems that convert photoinduced state transition to microflow of DNA fluid capable of mechanical manipulation” since there is no directionality in the process. In fact, no idea is given about how to use this finding to obtain “technologies for remote-controlled fluid manipulation [...] such as DNA-fueled transport, artificial micromuscles, and microstructure compaction.” The claim “the photoresponsive DNA fluid hydrodynamically performed mechanical actions upon the nonphotoresponsive DNA fluid by transducing light energy” is incorrect since the energy used for melting is just a fluidization and not converted into directional motion.

Connected to this point is the lack in the manuscript of any quantitative evaluation of such energy conversion claims. The authors talk about “enthalpy fine-tuning in SE binding affinity”. How can they claim such a thing without quantitative estimates of energies and ranges?

For this work to be appreciated and generate an impact, the authors should first describe in detail the thermodynamics of the azo-conjugated Y DNA under irradiation both in temperature and in time. And answer questions such as: with no Vis irradiation, what is the recovery time of the gel? Are the authors sure the Vis irradiation makes a difference? Answering this question would require studying the kinetics at constant temperature. Is the gel-liquid transition sharp? If not, how wide are such temperature range? What is the diffusivity of the DNA domains with no constraint? What the surface tension (is it measurable)?

In other words: to be of impact this work should offer the reader a clear characterization of the azo-conjugated DNA before focusing on the heterogenous mixture, which can be better appreciated if the azo response is clear.

Also, how the mechanism of “fetching” depending on surface tension? (“the subsequently emerging surface tension led to drastic compaction of the “cargo” DNA gel particles”). I assume the surface tension the authors refer to is the one between liquid and gas state of Y azo DNA constructs. If not they should state more clearly what they mean. Aren’t the “cargo” domains already embedded in an azo-conjugated Y DNA gel? Thus, as the gel melts, I would expect them to be embedded in an azo-conjugated Y DNA liquid, with no surfaces involved, and thus not be exposed to the surface tension effect. Isn’t the density in the gel and liquid phase about the same? Again, these questions arise because the description of the pure system is so vague.

Finally, I noticed that the original paper on phase transition of DNA nanostars by Biffi and coworkers is not cited.

Version 1:

Reviewer comments:

Reviewer #2

(Remarks to the Author)

The paper was extensively revised, and the authors made a real effort to improve the paper.

One minor comment relates to the switchable transitions of the condensates defined by the authors to undergo between gel /liquid /dispersed states.

This reviewer feels that the term “dispersed” is inappropriate. It basically presents a fully dissociated state of the condensates constituents. The authors are suggested to exchange the term “dispersed” to “separated constituents” throughout the paper.

Otherwise, this is a nice and interesting paper that is recommended for publication in Nat. Communications.

(Remarks on code availability)

Reviewer #3

(Remarks to the Author)

[Note from the Editor: Reivewer #3 was asked to assess also the response given to reviewer #1.]

I inspected the revised version of the manuscript now entitled “Remote-controlled mechanical and directional motions of photoswitchable DNA condensates” by Udono et al. I find it much improved with respect to the previous version, and I am in favor of publishing it.

I must however say that the inflated style of abstract and introduction still annoys me. What is the purpose of the 1-3 requirements for “microfluidic actions”? It is clear that they have no part in the MLO example given right after in the same section. Also, what is described is not a real “photocontrollable locomotion” on the micron scale in the way everybody would desire it to be. I see no way in which these findings will turn into “programmable, remote-controlled microfluidic systems, including DNA-fueled selective transport, artificial micromuscles, and microstructure folding.” and I dislike science too full of hypes, in which the reader must navigate between claims to get the real core findings.

Nevertheless, since the findings are interesting, and since my duty is not to address the publication style (this is to the editors to decide), I give my green light to the publication.

(Remarks on code availability)

Dear Reviewers,

Below is attached our point-by-point response letter addressed to Reviewers#1, #2, and #3, including a list of changes made. For clarity, the sentences and words newly added in the revision are colored in red, and those deleted in gray. The sentences related to the major comments and suggestions from Reviewers are highlighted in yellow backgrounds when supplemented with more detailed descriptions in Supplementary Information.

Reviewer#1

Thank you for your sincere and careful reading of the manuscript.

We have revised the manuscript to address your suggestions and questions as raised below, which improve the manuscript drastically.

Major updates in this revision round are:

- Our claims have toned down to address the language issue.
- Quantitative reinforcement of our previous claims in new Fig. 2d,e (basic characterization of the photo-switched phase states) and new Extended Data Fig. 3c,d (experimental evaluation of enthalpy difference ΔH in azo-modified SE binding)
- Demonstration of directional motion of locally photo-switched DNA liquid condensates (new Fig. 7), with numerical modeling (Fig. 8, Extended Data Fig. 4)

Major

Our responses

1. The manuscript describes a molecular system in which the properties of DNA-based materials are photo-switched using the already widely and frequently reported azobene-modified sticky-end strategy.

(1) Indeed, we agree that photo-switched DNA materials, previously shown in some existing works (*e.g.*, Le Ny et al., Photoreversible DNA Condensation Using Light-Responsive Surfactants. *Journal of the*

Particularly, (1) DNA materials are photo-switched between gel and liquid states as previously published including in references cited in the manuscript. (2) The uses of the terms “non-equilibrium microflow”, “microfluidic mechanical action”, “cargo and transporter”, “gather”, and “fetch” seem like spin or hype or marketing language rather than some new, important science or engineering. Simply put, thermal heating plus steric (trans Azo) melting of sticky-end adhesion in DNA gels results in enhanced diffusion in liquid states, much of the rest of the rhetoric seems like window-dressing. While (3) it is somewhat difficult to see the morphology changes of the blue/green “fluid” in the still images (for example Fig 4d, Fig 5), the enhanced diffusion during gel melting is much more obvious in the Supplemental videos.

American Chemical Society **128**, 6400–6408 (2006); Agarwal, S. et al. "Light-controlled growth of DNA organelles in synthetic cells." *Interface Focus* 13.5 (2023): 20230017.; Zakrevskyy, Yuriy, et al. "DNA compaction by azobenzene-containing surfactant." *Physical Review E—Statistical, Nonlinear, and Soft Matter Physics* 84.2 (2011): 021909., etc.), have been based on azobenzene (azo) or its relatives.

However, the primary focus of our paper is on:

- Mechanical actions (new Figs. 4–6) and directional motions (Figs. 7, 8, Extended Data Fig. 4) of phase-separated DNA condensates
- Thermodynamic quantification of ΔH in the azo-modified SEs (Extended Data Figs. 3c,d).

None of these aspects, we believe, have been considered, because most of the existing works have been limited to dynamic reconfiguration and size adjustment of azo-modified DNA systems upon photoswitching, with or without phase-state change, accompanied by no mechanical actions as demonstrated herein.

This is also the case with non-DNA material, such as peptides (Renner, Christian, and Luis Moroder. "Azobenzene as conformational switch in model peptides." *ChemBioChem* 7.6 (2006): 868-878; Mart, Robert J., and Rudolf K. Allemann. "Azobenzene photocontrol of peptides and proteins." *Chem. Commun.* 52.83 (2016): 12262-12277., etc.).

The photo-switched phase-state change is just a starting point as a basic means to achieving our goals to create multi-modal mechanical actions.

Also, in terms of remote-controlled microfluidics, our photo-activated directional motion of DNA liquid condensates is reasonable within a confined environment, such as biological or synthetic cells. A recent paper featuring remote-controlled droplets capable of directional motion (Chaaban et al. “Omnidirectional droplet propulsion on surfaces with a Pac-Man coalescence mechanism”, *Phys. Rev. Fluids*, 2020) is enabled by an ink-jet printing technology. This external-device dependency for the material supply is not amenable to such microfluidic remote control within confinements.

In our response, the title of the revised manuscript has been updated with: “Remote-controlled mechanical and directional motions of photoswitchable DNA condensates”.

Note that ‘photoinduced phase transition’ has been removed from the original one, since we claim no novelty in this phase-transition process itself.

We have cited the above papers (Agarwal, S et al.; Zakrevskyy, Y et al.; Renner et al.; Mart et al.; Chaaban et al.), previously uncited in the previous manuscript.

To emphasize this point, we have also corrected Abstract as:

“Leveraging base-sequence programmability, spatially coupled orthogonal DNA condensates with divergent photoresponsive capabilities perform multi-modal mechanical actions that depend on azobenzene insertion sites in the SE, including switching flows radially expanding and converging under photoswitching.”

In Introduction:

(p.5, middle) “Moreover, photoswitchable materials reported thus far, using DNA or other biomolecules, have achieved no significant mechanical actions, with a primary focus on controlled compaction and phase change”

(2) Our use of the words, which Reviewer describes as “window dressing”, is aimed at minimizing potential confusion of the non-expert readers, who might have to follow our DNA designs and multiple flow modes. We imagined that these words were able to give their meanings clearly in an intuitive and concise manner.

To correct Reviewer’s misunderstanding, we have responded as follows by replacing some of the words, which we think sounded like marketing words, with more neutral ones:

- “non-equilibrium microflow” (p.30, in Conclusions) has remained un-corrected, because the our reported microfluidic phenomena reflect non-equilibrium aspect of the thermodynamic changes of the systems as discussed in Extended Data Fig. 3c,d (quantitative

estimation of ΔH in azo-modified SEs). Indeed, our data set includes various time-dependent data (ex. Fig. 2e, Fig. 5, Fig. 7h, Fig. 8c). These results reflect the thermodynamic changes at molecular levels and the consequent non-equilibrium processes. Hence, the “non-equilibrium microflow” phrase is not a window dressing.

- “microfluidic mechanical actions” (p. 3, bottom): This phrase has been removed from previous Fig.5 caption; another one in Introduction when raising a general idea has also been removed as “establishing remote-controlled microfluidic _____ actions requires...”.
- “transporter” has been deleted
- In Fig. 5e, “gather” has been replaced with “collapse” frequently encountered in scientific papers; “fetch” (Fig. 5g) with “collect”.

(3) Thank you for carefully checking the figures.

To address the confusion, we brushed them up with the following:

- (In the previous Fig. 4d) We are not certain if azo insertion site changes the initial morphology of the resulting condensates of cross-linked DNAs; however, the figure is not aimed at showing this. To clarify this point, we have:
 - Deleted the previous Fig. 4d of the snapshots of the initial states
 - Added representative snapshots capturing the initial morphologies for different Azo insertion sites with fluorescence microscopy (Supplementary Fig. 4), where photo-responsive

	DNA condensates are in green and those non-responsive in blue. It is repeated that we have no intention of claiming that varying the azo insertion site can affect the morphologies. ➤ In relation to this point, we have made a concluding remark for Supplementary Fig. 4 as (p.14, lower) “Various adhered DNA condensates in a gel state (Supplementary Fig. 4) show that the cross-linker DNA could spatially couple the different photoinduced phase behaviors within one system.”  - (In new Fig. 5e) An Illustrator layer of blue particles moved up on top of green liquified DNA layer - (New Fig. 5d,f,h) Yellow dashed-line curves have been added as an eye guide to mark the outermost periphery of the advancing Y_0 condensate. - Previous Fig. 5i (cyclic motion of a Y_0 condensate) have been cut, and merged with previous Supplementary Fig. 2, which was moved to the revised main manuscript as Extended Data Fig. 2, including a kymograph that tracks the alternating expansion (UV) and convergence (Vis).
2. (1) Authors should define “reentrancy” within this context. This reviewer does not appreciate the phrase (2) “transient non-equilibrium	(1) Thank you for pointing this out. We agree that “reentrancy” sounds unfamiliar to most of the readers. We used “reentrancy” to highlight

state”, because it appears to refer to external perturbation of the system (e.g., irradiation) followed by re-establishment of equilibrium. The unperturbed systems are always tending toward thermodynamic equilibrium.

temperature profile of flow mobility, with a distinct peak in the intermediate temperature ranges accompanied a decrease in the lower and higher temperature ranges.

For clarity, we have:

- Deleted reentrancy/reentrant-related description
- Cut less essential sentences that seemed redundant “(pp.18–19) The single-peak profiles of... underlying mechanism of the reentrant flow-mobility behavior”
- Re-phrased “reentrancy” e.g., “Temperature-dependent flow mobility...” (new Fig. 6 caption, p.52) and (p.19, middle)“Thermodynamically, the observed reentrancy peak shift (between $Y_{3x5}/L_0/Y_0$ and $Y_{2x1x5}/L_0/Y_0$) and sign inversion (between $Y_{3x5}/L_0/Y_0$ and $Y_{1x7}/L_0/Y_0$) in...”

(2) We used “non-equilibrium” to refer to the re-establishment process toward equilibrium in response to external perturbations, as Reviewer points out. However, this study features the quantitative data and discussion about ΔH in SE binding as thermodynamic mechanism behind the peak shift/inversion in the flow mobility. This quantitative experimental estimation has been largely ignored in DNA condensate studies. To highlight the irradiation effect on the thermodynamic properties and consequent non-equilibrium process, we are convinced that the ‘non-equilibrium’ aspect should be reasonably dealt with. Indeed, our data set includes various time-dependent data (ex. Fig. 2e, Fig. 5, Fig.

	7h, Fig. 8c). These results relate to the thermodynamic changes at molecular levels and the consequent non-equilibrium processes. Hence, the “non-equilibrium” highlights the macroscopic consequences of these hydrodynamic changes at molecular levels. Nevertheless, in this context, ‘transient’ sounds redundant beside the similar meaning to ‘non-equilibrium’. To address this issue, we have:  - Added quantitative evaluation in ΔH in SE (new Extended Data Fig. 7c,d), which is discussed in detail in Supplementary Texts - (Introduction, p.6, lower) “With quantitative experimental validation of the photoswitched ... by DNA microflow that depends on the Azo insertion site.”. - Deleted ‘transient’ from (Conclusions, p. 30) “Photoswitching SE binding state resulting in transient non-equilibrium microflow of DNA, ...”
3. Melting of the gel proceeds well (Fig 1c), but reformation of the gel is very inefficient (Fig 1e). This appears to be due to dilution of the Y-motifs (or individual oligonucleotides) in surrounding buffer following melting of the gel. Individual sticky-end adhesions are shown to function as planned (electrophoresis results Fig 2b), but large scale gel formation may require a thermal anneal. This point is glossed over in the Results and Discussion with the statement	We agree that gel reformation in additional Vis irradiation as shown in the previous Fig. 2b(i) is not efficient. This arises from the isothermal conditions. However, our focus is on the reversibility of phase-state change, not on its efficiency. We have no intention of concealing this inefficiency. As discussed above in this sheet, we claim no novelty in the phase transition using azobenzene “already widely and frequently reported” in sticky end strategies.

“prescribed annealing process caused a set of three single-stranded DNAs (ssDNAs) to self-assemble into a Y-motif”. The gel state forms efficiently via thermal anneal, but not so well during VIS irradiation.	Thus, we present the data concerning Vis-induced re-condensation in Supplementary Fig. 2 about size distribution as a function of Vis irradiation upon UV-dispersed DNA. The data show that extended Vis irradiation (>20 min) after UV-to-Vis switching did not improve this inefficiency. In the revised manuscript, we have added:  - (p. 9, lower) A statement about this reformation inefficiency “We note that ... by obtaining the size distributions of recondensed DNA particles as a function of Vis irradiation time (Supplementary Texts, Supplementary Fig. 2).” - (Supplementary Text, p. 8, “3. Reconstruction of DNA condensate with UV-to-Vis switching”) Discussion on this low efficiency “We have shown the reversibility of the photocontrolled phase state...and counteracted the recondensation process.”
4. The authors use the term “DNA fluid” to indicate both the gel and liquid states. This seems strange. Fluids are typically liquids (or gases) (i.e., fluids flow). Why are the gel states also referred to as fluids? The term “fluid” is never defined in a concise and satisfactory way. Phrases like “two immiscible DNA fluids” used when referring to two gel particles seems odd. Yes, diffusion is slow (but non-zero) in gel states and faster (but finite) in liquid states, but this terminology labels the behavior as if it were absolute, black and white.	Thank you for valuable comments. In the original manuscript, we used ‘fluid’ as even gel-state condensate can be deformed and flow under a weak shear over a reasonably long time scale. Whether a specific physical entity can be characterized as fluid or not depends on time scale of interest. Also, the resulting microflow of DNA is in un-condensated state, which forced us to hesitate to label it as ‘condensate’, a much more typical noun for phase-separated system.

	Having said that, with higher priority for clarity, we have replaced ‘fluid’ with ‘condensate’, as suggested.
5. Comparing “Before UV” images in Fig 5c,d with those in Fig 5e,f it is not at all clear why the starting states are so different (i.e., condensed sphere in c,d and diffuse toroid in e,f) . Explain further.	We previously picked up those starting states, as they were able to capture well the distinct features of the different flow modes. For example, a toroid structure would display a UV-induced collapse behavior of the $Y_{1x7}/Y_0/Y_0$ condensate more markedly than the dense structure like Fig. 5d. We have added: (p.15, lower) “It is reminded that the initial states shown in Fig. 5 were selected specifically as configurations well suited to clearly display the distinctive flow modes. See other representative initial configurations in Supplementary Fig. 4.” (Supplementary Texts, p.11) “5. Initial configurations”
Minor	Our response
Correct typographical errors in Figures: Trasnporter (Fig 5), complimentary (Fig 4) (should be complementary), liquid (Fig 4)	Thank you for careful reading of the manuscript. We corrected and deleted these typos.

Reviewer#2

Thank you for professional comments and suggestions.

We have responded to these as below, which will largely improve the quality of this study.

Major updates in this revision round are:

- Our claims have toned down, to address the language issue that sounded un-supported and redundant.
- Quantitative reinforcement of our previous claims in new Fig. 2d,e (basic characterization of the photo-switched phase states) and Extended Data Fig. 3c,d (experimental evaluation of enthalpy difference ΔH in SE binding)
- Demonstration of directional motion of locally photo-switched DNA liquid condensates (new Fig. 7), with numerical modeling (Fig. 8, Extended Data Fig. 4)

Major

Our responses

1. The entire study of light-induced crosslinking of the Y-shaped structures includes the coupled effect of light/temperature on the crosslinking. The authors support the crosslinking events by a molecular level system presented in (1) Figure 2 that includes only light as the crosslinking motive. The temperature used in this experiment is not given, and (2) then the relevance of this model system to the liquid DNA assembly is questionable. The authors should discuss and explain this point. (3) Also, the results presented in

(1) In addition to the temperature in Method section, already given in the previous manuscript, we have added the temperatures in the main body:
- Extended Data Fig. 1a (PAGE, previous Fig. 2) caption, “4°C”

(2) We agree that the lack of temperature information in PAGE section might reduce the relevancy of the single-SE motif to the DNA condensate as a model system. Also, we imagine that even adding this in the revised manuscript may not address well Reviewer’s comment, as no other temperatures were not considered in the PAGE like other experimental section including multiple temperature ranges.

As a model system for PAGE experiments, however, we had no other choice but this ‘single-SE’ motif, as ‘triple-SE’ motif would form condensates and prevent distinct binary

Figure 2(C) show very small differences. Error bars are missing to justify the conclusions.	band formation (They are trapped in the loading well inlets). Furthermore, this single SE motif, with significantly lower binding strength than the triple SE motif, yielded weak band intensities at room temperature. We thus determined that PAGE should be carried out at 4°C for distinctive band formation. To comment on this, we have modified as follows:  - (p.10, middle) “Here, we used Y-motifs with a single SE to highlight the SE binding states while preventing condensation. As illustrated in Extended Data Fig. 1a, the single-SE motifs adopt only two states, the associated (dimer) and dissociated (monomer) states, corresponding to the upper and lower bands in a lane, respectively...” - (p. 37, top) “This low-temperature condition was preferred because PAGE results performed at room temperature (RT) showed less distinct band images.” (3) We have added the error bars in new Extended Data Fig. 1c, where $n = 3$.
2. The phase transition relies on temporal confocal fluorescence imaging and lacks quantitative evaluation. Supporting the conclusion by quantitative temporal viscosity changes, turbidity and light scattering experiment should be provided.	We highly appreciate these professional suggestions, and have responded by adding FRAP experiments on DNA condensates (new Fig. 2d,e; Supplementary Fig. 3). In FRAP, we inverted the typical procedure due to photo-sensitivity of the condensate, i.e., bleaching, irradiation, and then fluorescence recovery. We have described this as follows:  - (p.11, top) “The irradiation-induced fluidity alterations were indicated as a consequence of the phase behaviors of DNA condensates...These FRAP results confirm the photoreversibility in the motif diffusion properties of condensates.” - Supplementary Information 4. “FRAP experiments for photoresponsive DNA”

	Meanwhile, we considered but gave up performing OD measurement of condensate solution, as suggested by Reviewer, due to the photo-sensitivity of condensate. Furthermore, we tried DLS experiments for probing viscosity or elastic modulus using gel condensate solution confined within a cuvette. However, no significant signals were detected due to insufficient DNA concentration. Indeed, increasing DNA concentration that is enough to establish space-spanning gel structures, as shown in Biffi et al. (2013), might give significant signals. However, such condition would be a huge diversion from the optimal conditions suited to form the phase-separated condensates investigated here. The light scattering methods also affect the phase-state of the photo-responsive condensates.
3. (1) The illumination time-intervals of the model system shown in Figure 2 proceed on a time-scale of minutes. On the other hand, the dynamic mobility experiment shown in Figure 5 and Figure 6 includes irradiation time intervals of seconds (and relatively high temperatures). (2) In general, a counter effect of thermal cis-to-trans isomerization of the azobenzene takes place, and the effect of this process on the phase transition is ignored. A solid discussion on the effects of the thermal cis-to-trans transition of the azobenzene unit on the phase transition processes should be provided.	(1) Indeed, we used different time scales for photo-induced reaction. In phase-transition assays (new Fig. 2a,d) and PAGE experiments (new Extended Data Fig. 1, previous Fig. 2), we preferred minute-to-hour scales. This was because we used weak UV intensity. A strong irradiation would produce drastic changes and even unwanted disturbance around the condensates, making it harder to achieve gradual changes in SE binding. We have added:  - Irradiation intensities used for UV and Vis in Figure captions, where irradiation assays were conducted - Additional remarks about the above explanation:  - (p.8, middle) “The constructed DNA condensates were irradiated for 2 h with sufficiently weak UV to capture their gel-to-liquid morphological changes while reducing disturbance flows around the condensates.”

- (p.10, lower) “The samples to be loaded in the lanes were irradiated by 6-min UV, followed by 6-min Vis”

Meanwhile, in the microflow experiments (new Figs. 5–), we preferred a timescale of seconds at strong irradiation intensities that were better for generating microflows.

We have added:

(p.15, upper) “The constructed DNA condensates of cross-linked motifs were irradiated with collimated excitation light ... significantly reduced the reaction timescales from minutes to seconds.”

(2) Thank you for the professional comments, which we consider to be valuable for more rigorous evaluation and solid support.

In new Fig. 2b(ii), we performed a control experiment in dark conditions for a longer time (5 h incubation) at 53°C, the same temperature as that of Vis reversing experiment, to check the thermal *cis-to-trans* effect. But we found no significant contribution to the recondensation process. We have added this data and discussion:

- (p.9, lower) “To scrutinize the effects of thermal *cis-to-trans* isomerization of Azo molecules on the recondensation, we performed a control experiment in dark condition over an elongated time. Even after 5-h Vis irradiation, no distinct condensate growth was available, suggesting that the thermal *cis-to-trans* Azo isomerization contributed negligibly to the recondensation process.”

Minor:	Our responses
- (1) Figure 5(e) the gathered Y-Motif should be colored blue. - Figure 6(a,b) scale bars should be added.	Thank you for careful reading. (1) We have corrected the Y-motif color by changing the layer of an illustration file. (2) 50-μm scale bars have been added to new Fig. 6a, b.

Reviewer#3

Thank you for valuable comments and constructive suggestions.

We have responded to these as below, which we believe will strengthen the quality of this study.

Major updates in this revision round are outlined:

- Our claims have toned down, to address the language issue that sounded un-supported and redundant.
- Quantitative reinforcement of our previous claims in new Fig. 2d,e (basic characterization of the photo-switched phase states) and Extended Fig 3. c,d (experimental evaluation of enthalpy difference ΔH in azo-modified SE binding)
- Demonstration of directional motion of locally photo-switched DNA liquid condensates (new Fig. 7), with numerical modeling (Fig. 8, Extended Data Fig. 4)

Major

Our responses

1. While the topic of light control of the aggregation state of a DNA hydrogel can be in principle interesting, the manuscript is, in my opinion, not worth publishing. The manuscript is bombastic, watering down the potentially interesting scientific findings within pages of pompous claims, such as “an exceptional technique for versatile, simple microflow manipulations with mode switchability” (there are tens of these).

As suggested, we have removed all those “pompous claims” to tone down the language in our previous manuscript. We agree that “versatile” sounds too strong, and have remove words and phrases including this word.

Also, we have cut the whole statements that came at the end of ‘Results and discussion’ section, which were full of seemingly un-supported and promotional comments and suggestions, including “an exceptional technique for versatile, simple microflow manipulations with mode switchability” and “The use of other photoreactive molecules could not offer such a facile method for coupling multiple phase behaviors, and hence annoy researchers and engineers due to the cumbersome manufacturing processes”. See the grayed-out sentences in pp. 29–30.

	Further, we have replaced apparently hyperbole words and phrases to more neutral ones, such “fetch” to “collect” and “gather” to “collapse”. Meanwhile, we have added reinforcing data on the basic characterization of photo-controlled phase transition and consequent property change of DNA condensates (FRAP experiments, new Fig. 2d,e) and quantitative estimation of thermodynamic parameters in SE binding ΔH (Extended Data Fig. 3c,d). Overall, we believe that a large deletion of pompous claims and exaggerated phrases and an addition of new supporting data and findings will enhance the quality of this paper.
2. The authors show that upon irradiation the DNA gel transforms into either a liquid (see question below), or in a “gas”. This is interesting but it is not a “energy- transducing systems that convert photoinduced state transition to microflow of DNA fluid capable of mechanical manipulation” since there is no directionality in the process. In fact, no idea is given about how to use this finding to obtain “technologies for remote-controlled fluid manipulation [...] such as DNA-fueled transport, artificial micromuscles, and microstructure compaction.” The claim “the photoresponsive DNA fluid hydrodynamically performed mechanical actions upon the nonphotoresponsive DNA fluid by transducing light energy” is	We greatly appreciate Reviewer for the professional and constructive suggestions. Indeed, we agree that our initial presentation of photo-controlled DNA microflow on a fixed position in the previous manuscript lacked its further exploitation as energy-transducing system. In the revised manuscript, we have additionally explored the potential of photo-controlled DNA condensates by replacing the isotropic global photoswitching with localized anisotropic photoswitching. In new Fig. 7, we have demonstrated directional motion of a DNA liquid condensate under localized UV–Vis switching. We have found jellyfish-like swimming, referred to as “push-swimming” at low switching frequencies

incorrect since the energy used for melting is just a fluidization and not converted into directional motion.

and Pac-Man like swimming, called “pull-swimming” at higher but intermediate switching frequencies.

We have added:

(p. 22, top–p. 23, upper) “The irradiated area used above was radially symmetric, resulting in no net directional displacement of the liquid condensates...At much higher switching frequencies, we only observed minimum displacements, where no DNA microflow discharged from the condensate.”

Further, coupled with cross-linker system, already shown in the previous manuscript, these swimming behaviors have achieved cargo transport, with non-photoresponsive DNA gel condensates serving as cargo. This demonstrations has been added:

(p. 23, top–p. 24, top) “Further, we employed these two types of condensate swimming for cargo transport to exemplify their great potential as a microfluidic tool in a broad range of applications...a similar rolling behavior was detected in some small irregularities being rolled up on the surface (Supplementary Movie 8).”

To gain a basic understanding of the underlying mechanisms behind these swimming modes, we have carried out numerical simulations using a simplified model that featured mass and energy transfers between the ‘Liquid’ and ‘Dispersed’ states (new Fig. 8, Supplementary Information for detailed model description). We argue that photo-switching frequency regulates a balance between the energy-dissipative interaction (dominant in

	push-swimming) and the energy-exchanging interaction (dominant in pull-swimming). These statements have been made: (p. 24, top–p. 27, lower) “For a greater understanding of the underpinning mechanisms, we performed numerical simulations to ... Thus, the switching frequency f is key to adjusting the weight of the energy-dissipating and energy-exchanging contributions in the mass and energy transport.” This basic understanding further allowed a rough estimation of a threshold for the pull-swimming regime (more energy-exchanging dominant): (p. 27, bottom–p.28, middle) “Assuming that ...
3. Connected to this point is the lack in the manuscript of any quantitative evaluation of such energy conversion claims. The authors talk about “enthalpy fine-tuning in SE binding affinity”. How can they claim such a thing without quantitative estimates of energies and ranges?	We highly appreciate Reviewer for this beneficial suggestion. Indeed, we missed quantitative estimation of enthalpy difference in SE between the bound and un-bound states ΔH, considering the limited availability of experimental approaches toward thermodynamic properties of DNA condensate. In the revised manuscript, we have added quantitative estimation of ΔH in azo-modified SEs with experimental support (new Extended Data Fig. 3c,d), in relation to our previous argument of “enthalpy fine-tuning in SE binding affinity” (pp.19–20), supplemented with thermodynamic discussion in Supplementary Information (p. 16–p.18). There are no established theoretical models predicting the thermodynamic parameters (ΔH, ΔS) for azo-inserted DNA sequences. Most of those well-

established, such as nearest-neighbor methods, are applicable but limited to non-modified sequences due to a limited list of the related parameters. Instead, we measured dissolving temperature T_D of DNA condensates with various azo-free SEs, whose thermodynamic values can be simulated easily.

Then, we have shown that experimentally achievable T_D of condensates and thermodynamic parameter ΔH in the corresponding Azo-free SEs are monotonically related (new Supplementary Fig. 7), and have also derived a fitting curve (Extended Data Fig.3c) to be applied for quantitative estimation of ΔH of azo-modified SEs.

Then, taking together T_D for azo-modified SEs (Fig. 3), already measured in the first draft, with the $\Delta H-T_D$ fitting curve (new Extended Data Fig. 3c), we have made estimation of various ΔH for *trans/cis* azo-modified SEs (Extended Data Fig.3d). This quantitative estimation has justified our previous argument that Azo insertion site in the SE fine-tunes ΔH in the SE, and hence the flow mobility in the temperature dependence of the resulting DNA microflow is specified as a result of the photoswitched gap of $\Delta(\Delta H)$.

We have added the new quantitative data and related explanations in (p. 19, middle–p. 20, top) with more detailed discussions in Supplementary Information (pp. 16–18).

4. For this work to be appreciated and generate an impact, the authors should first describe in detail the thermodynamics of the azo-conjugated Y DNA under irradiation both in temperature and in time. And answer questions such as: (1) with no Vis irradiation, what is the recovery time of the gel? (2) Are the authors sure the Vis irradiation makes a difference? Answering this question would require studying the kinetics at constant temperature. (3) Is the gel-liquid transition sharp? If not, how wide are such temperature range? (4) What is the diffusivity of the DNA domains with no constraint? (5) What the surface tension (is it measurable)? (6) In other words: to be of impact this work should offer the reader a clear characterization of the azo-conjugated DNA before focusing

We have addressed Reviewer's professional comments as follows:

(1) With no Vis irradiation, we have carried out a control experiment at 53°C (constant) to evaluate the 'recovery time' of UV-dispersed condensates. In new Fig. 2b(i), we observed noticeable nucleation within 1 min of Vis irradiation, followed by significant size growth within 20 min Vis irradiation, whereas no emergence of distinct condensate particles over a 5h incubation in dark condition (Supplementary Fig. 2b).

(p. 9, upper–p. 9, lower) “Next, Fig. 2b shows a Vis-induced dispersed-to-liquid transition.... for time-lapse confocal microscopy imaging of Vis-induced recondensation at 56°C”

(2) Thus, Authors are sure of the Vis effect, since we performed a control experiment at a fixed temperature of 53°C (new Fig. 2b(ii)).

(3) Before answering this question, we have added some more information on the UV-induced gel-to-liquid transition:

In new Fig. 2a, gel-to-liquid transition is shown with control experiments. Within 30 min of UV irradiation, the initially gel-state condensate morphologically transformed from sponge-like to more smooth structure with smaller size, and finally rounded up in 120 min.

on the heterogenous mixture, which can be better appreciated if the azo response is clear.

Ideally, more smaller time intervals (e.g, 1 min) would be better to probe the morphological changes of the condensate. But due to the high photo-sensitivity of azo-modified DNA condensate, an interval was set to 30 min to minimize the Vis effect from the bright field observation. Nevertheless, considering the large compaction over 30 min, distinct gel-to-liquid was likely within ~10 min.

Of course, whether “transition is sharp or not” depends on irradiation intensity. The light intensity of a xenon lamp used was weaker than that used for microflow experiments (in Figs. 5–) using hydrogen light ejected from a microscopy objective lens.

The above statements have made as:

(p.8, middle–bottom) “The constructed DNA condensates were irradiated for 2 h with sufficiently weak UV to capture their gel-to-liquid morphological changes while minimizing disturbance flows around the condensates. ...The significant changes in the size and morphology indicate that the condensates achieved a significant gel-to-liquid phase transition within ~10 min of UV irradiation”

(p.15, upper) “The constructed DNA condensates of cross-linked motifs were irradiated ... reduced the reaction timescales from minutes to seconds.”

Back to answering Reviewer’s question, we find it hard to answer that “the gel-to-liquid phase transition was sharp”. At least, the molecular-level isomerization is sharp enough, as shown by Asanuma et al. However, at the

condensate level, there is a characteristic time for the molecular-level reactions to translate to the macroscopic level. In the observed UV induced melting, we estimate ~10 min or so with the Vis intensities used, but whether it is sharp or not depends on the exerted irradiation intensity, gel size etc. In the revised manuscript, instead, we simply stated the UV-caused melting as (p. 8, bottom) “The UV-induced rounding-up of condensates strongly suggests...rheological discussions are available in Supplementary Texts and Supplementary Fig. 5a)”

For the question “how wide are such temperature range”, the UV-induced melting was carried out at the fixed temperature, 46°C, with no thermal cycling.

We have clarified this as:

(p.8, middle) “at a constant temperature of 46°C” for UV melting and also (p. 9, upper) “at a constant temperature of 53°C” for Vis recondensation experiments.

(4) We experimentally evaluated the diffusivity of constituting Y DNA motifs in the gel-state condensate (35°C) with FRAP experiments (new Fig. 2d,e).

Briefly, UV led to higher diffusivity, and switching Vis resulted in a decrease in the original diffusivity level.

- (p.11, top–bottom) “The irradiation-induced fluidity alterations were indicated as a result of the phase behaviors of DNA condensates... These FRAP results suggest the photoreversibility in the motif diffusion properties of condensates.”
- (Supplementary Information, pp. 9–10) The detailed protocol and analysis method of FRAP

(5) In the revised Main manuscript and Supplementary Information, we have added fundamental briefing on the “surface tension” for a better understanding of the “surface tension”-related morphological changes observed here. This fundamental description has been given at:

- (p. 8, bottom), “The UV-induced rounding-up of...(more detailed discussions are available in Supplementary Texts and Supplementary Fig. 5a)”
- Supplementary texts (p. 12, “7. Surface stress effects in UV-induced phase-state changes”).

At an interface separating two immiscible phases, there is a competition between surface stress S (interfacial force per unit length acting on a slightly deformed region as a manifestation of surface energy γ) and condensate elasticity E (Style et al., 2017). When the condensate is in a liquid state, the surface energy at the liquid–liquid interface is equal to the surface stress ($\gamma = S$). This corresponds to surface tension, a physical property previously studied of phase-separated “liquid” condensates (Feric et al., 2017; Sato and Takinoue, 2023; Gouveia et al., 2022). When it is in a

gel state establishing a (soft) solid–liquid interface, $\gamma \neq S$ (Andreotti and Snoeijer 2016; Shuttleworth 1950). As the photoirradiated phase state change involves both gel and liquid states, we prefer to use surface “stress” S throughout the revised manuscript to minimize potential confusion, instead of surface “tension”.

Whereas at length scales $l > l_c (= S/E)$ surface effects are negligible, only at length scales $l < l_c$, surface stress effects dominate, inducing an interfacial deformation (Style et al. 2017; Bico, Reyssat, and Roma, 2018). Initially, in the solid gel-state condensate, E fully outweighs S , where most of the length scales fall within $l > l_c$, accompanied by no condensate deformation. With an increase in the fluidity (*i.e.*, smaller E), S outstrips E . This leads to a wider range of length scales satisfying $l > l_c$ and hence a noticeable condensate deformation, where the surface energy acts to minimize the surface-to-volume ratio, giving rise to a rounding-up of the irregularly shaped condensate. Hence, the observed rounding-up of the UV-irradiated condensate suggests a UV-induced gel-to-liquid phase change and a subsequent increase in the fluidity.

Also, the observed fluidity is a macroscopic manifestation of enhanced motif reshuffling, where the building blocks interchangeably dissociate and dissociate, resulting in an increased fluidity (Cardellini et al., 2023). Overall, in relation to the UV-induced rounding-up of condensate, we have explained the following:

(p. 8, bottom) “The UV-induced rounding-up of condensates strongly suggests an increase in their fluidity as a consequence of the minimization of a surface-to-volume ratio in the gel-to-liquid phase-transitioning condensates (rheological discussions are available in Supplementary Texts and Supplementary Fig. 5a)”

For a graphical understanding of the above explanation, we have attached Supplementary Fig. 5a.

Back to Reviewer’s question as to the measurability of surface stress, in principle, surface “tension” of DNA liquid condensates is measurable via aspect ratio analysis of two coalescing liquid condensates, coupled with an equation of $f(t) = 1 + A\exp(-t/\tau_f)$, where t : time, A : fitting parameter, and τ_f : characteristic fusion time (Gouveia, B et al. (2022). Capillary forces generated by biomolecular condensates. *Nature*, 609(7926), 255-264.; Sato, Y., & Takinoue, M. (2023). Sequence-dependent fusion dynamics and physical properties of DNA droplets. *Nanoscale Advances*, 5(7), 1919-1925.). This method is a typical approach to the rheological evaluation of phase-separated liquid condensate (Folkmann et al, Science, 2021 and many other). However, this study is not focused on the measurement of the physical properties, as the proof-of-concept demonstration was just enough.

Importantly, this method is limited to the liquid states, as the gel-state condensates would conceal surface “stress” by the large elasticity.

Moreover, such measurement would require a lot of time-sequential data over an elongated observation, which can counteract UV effect on the phase-state change of DNA condensate. Therefore, surface “tension” measurements are not realistic.

(6) As described above, we have added the control experiments for UV and Vis irradiation (new Fig. 2a,b), together with the FRAP experiments (new Fig. 2d,e), using non-cross-linked Y_{3x5} in the manuscript. Indeed, the heterogenous mixture using cross-linker DNA was aimed at demonstrating sequence-specific control of photo-responsiveness.

At the same time, for the basic characterization of photo-generated microflow mobility, this mixture system was crucial, as the non-photoresponsive DNA cross-linked with the photo-responsive DNA serves as a good tracer. Other microbeads would not favor to be fully embedded in condensate DNA, thereby disqualified as a good tracer.

We have made this comment:

(p. 14, bottom–p.15, top) “Another advantage of the cross-linked motif design was the traceability of the photoresponsive DNA microflow in the quantification of its mobility, as shown below...Commercially available microbeads, which would not favor to be fully embedded within the condensate, were disqualified for the tracing assays.”

	Further, as discussed earlier here, for the basic characterization of the phase-state change of the photoswitchable condensates, FRAP is virtually the only way toward quantitative estimation of the fluidity as a key indicator of the photoinduced phase-state changes. Thus, we have related the fluidity with FRAP as: (p.11, upper–bottom) “The irradiation-induced fluidity alterations (Figs. 2a, b) were indicated as a result of the phase behaviors of DNA condensates...Taken together, these data provide experimental support that the photogenerated SE binding state and consequent motif mobility determines the phase state and fluidity of DNA condensates.”
5. (1) Also, how the mechanism of “fetching” depending on surface tension? (“the subsequently emerging surface tension led to drastic compaction of the “cargo” DNA gel particles”). I assume the surface tension the authors refer to is the one between liquid and gas state of Y azo DNA constructs. If not they should state more clearly what they mean. (2) Aren’t the “cargo” domains already	(1) Thank you for valuable comments. We agree that the previous description was not fully descriptive of the observed dynamics and its underlying mechanism of what Reviewer mentions. Based on the sentence that is echoed from the previous manuscript, Authors understand that “fetching” in the question refers to new Fig. 5e (labelled as “collapse” motion, previously denoted as “gather” motion), rather than new Fig. 5g (“spread & collect” motion, previously labeled as “spread & fetch” motion). Then, we can address Reviewer’s question as to how surface tension worked in “collapse” motion as follows. The observed “collapse” behavior included (α) large compaction of the overall structure and (β) self-folding of the non-photoresponsive Y_0 gel

embedded in an azo-conjugated Y DNA gel? (3) Thus, as the gel melts, I would expect them to be embedded in an azo-conjugated Y DNA liquid, with no surfaces involved, and thus not be exposed to the surface tension effect. (4) Isn't the density in the gel and liquid phase about the same? Again, these questions arise because the description of the pure system is so vague.

structure, based on a closer inspection of the Fig. 5f and Supplementary Movie 4. Below, we explain (α) and (β) as elastocapillary effects (Bico et al., 2018) in what follows.

In general, surface stress S (which Reviewer refers to as “surface tension”) becomes prominent with an increase in the relative significance of surface to bulk elastic stress, as introduced above. This S acts on an interface of DNA and surrounding buffer, as illustrated in Sheet Fig. 1.

Sheet Fig. 1 Surface stress at DNA–Water interface

Next, in the context of heterogeneous DNA, an additional phase of DNA “B” introduces two more interfaces as in **Sheet Fig. 2**, as far as DNA “B” is not fully enclosed in DNA “A”. Here, surface stresses are relevant at A–B, A–W, and B–W interfaces. Importantly, these surface stresses are balanced at the three-phase contact line, where the three surface stresses meet.

Sheet Fig. 2 Scenario 1 of triple-phase system, where DNA “B” not fully enclosed in “A”.

If DNA “B” is fully enclosed within DNA “A” phase, the system becomes the following:

Sheet Fig. 3 Scenario 2 of triplet-phase system, where DNA “B” is fully enclosed in “A”.

In this case, surface stresses are relevant at A–B and A–W.

This conceptual framework is applied to the observations (Fig. 5d,f,h; Supplementary Fig. 5b).

Initially, non-photoresponsive DNA gel particles (in blue) appeared to be interfaced with photoresponsive DNA gel particles (green) as well as with the surrounding buffer (water). And the UV-induced “collapse” behavior started with this gel-state condition, corresponding to scenario 1 (Sheet Fig. 2). In this initial condition, surface stresses $S_{1 \times 7 - W}$ (at $Y_{1 \times 7 - W}$ interface),

S_{1x7-0} (at $Y_{1x7}-Y_0$ interface) and S_{0-W} (at Y_0-W interface) remained suppressed by the larger gel-state elastic moduli E_{1x7} (of condensate Y_{1x7}) and E_0 of condensate Y_0 . With the UV-induced gel-to-liquid phase change of Y_{1x7} and increased fluidity, surface stresses S_{1x7-W} and S_{1x7-0} became prominent by outweighing E_{1x7} , similarly to the non-heterogenous system rounding up (Fig. 2a).

As S_{1x7-W} took effect at $Y_{1x7}-W$ interface, the $Y_{1x7}/L_0/Y_0$ mixed condensate favored more compacted state, as surface energy γ_{1x7-W} acted to minimize the surface-to-volume ratio.

Meanwhile, at the three-phase boundary ($W-Y_{1x7}-Y_0$), S_{1x7-0} and S_{1x7-W} jointly acted to change a force balance. Due to the binding affinity between Y_{1x7} and Y_0 (conferred by cross-linker L_0), Y_{1x7} phase preferred to maximize the $Y_{1x7}-Y_0$ interfacial area, a phenomenon known as wetting.

Then, at this step, elastocapillary effects came into play. Although surface stress S_{0-W} remained outstripped by elastic modulus E_0 , Y_0 gel condensate was still a highly deformable solid, unlike metal sheet. Consequently, the wetting allowed increasingly prominent surface stress S_{1x7-0} to fold the sparsely arranged Y_0 gel structure. This self-folding has been recently highlighted as elastocapillarity (Kwok et al., 2020; Bico et al., 2018). Similar examples include self-bending of microwires around a droplet and origami folding around a droplet, as introduced in **Sheet Fig. 4**.

Sheet Fig. 4 Self-folding and self-bending using surface tension. From Kwok et al. (2020).

Based on this understanding of the elastocapillary effects in the “collapse” behavior, we have explained as follows:

- (p.16, upper) “The compaction and self-folding process can be explained as a wetting process of the gel-to-liquid phase-transitioning DNA phase ($Y_{1 \times 7}$ condensates) interfaced with the deformable DNA gel

phase (Y_0 condensates), as discussed in Supplementary Texts (Supplementary Fig. 5b).”

- (Supplementary Information, pp. 12–14) “7. Surface stress effects in UV-induced phase-state changes”

For a graphical understanding of this description, we have added a step-by-step account in Supplementary Fig. 5b:

(2) As described above, the “cargo” gel particles, labeled as Y_0 , was NOT fully embedded within phase-transitioning Y_{3x5} , when the photoirradiation started. With melting, the Y_0 cargo was increasingly interfaced with Y_{3x5} due to increased wetting

(3) Thus, in the presence of the triplet “surface tensions”, the folding was available due to rebalancing between them, rather than no exposure of Y_0 gel particles to interfacial forces.

(4) Yes, about the same. The major difference is fluidity. Since the gel state contained lots of cavities, the apparent volume reduced as a result of UV-induced gel-to-liquid phase transition, giving the appearance of increase density. We have only negligible change in the volume occupied by the condensates. In fact, the surface energy acted to minimize the surface-to-volume ratio, leading to a beading up of the condensate into smaller size.

We have added:

	- (p. 8, bottom) “The UV-induced rounding-up of condensates strongly suggests... (rheological discussions are available in Supplementary Texts and Supplementary Fig. 5a)” - (Supplementary Information, p. 13, middle) “In addition, we believe that the density of the gel-to-liquid phase-transitioned condensates should have remained almost invariant throughout the photoinduced morphological change. The surface energy γ only acted to reduce the surface against the volume occupied by the state-transitioning condensates.”
Minor	Our response
Finally, I noticed that the original paper on phase transition of DNA nanostars by Biffi and coworkers is not cited.	We have cited the suggested paper at the asterisks: (p.4, bottom–p. 5, top) “Recently, DNA liquid condensates, micro-scale liquid condensate of DNA nanostructures with well-engineered sequences that emerges through phase separation, has drawn significant attention due to their sequence-specifically directed interaction and structural high programmability of structure, phase behaviors*, and physical properties**”

Response Letter (Round 2)

Reviewer #2——p. 2

Reviewer #3——p. 4

Reviewer #2

Thank you for positive comments and productive suggestion.

We have revised the manuscript to address your suggestion as raised below, which will really sophisticate the manuscript.

For clarity, the sentences and words updated in the revision (Main manuscript and Supplementary Information) are colored in red, and those deleted in gray.

Minor

The paper was extensively revised, and the authors made a real effort to improve the paper.

One minor comment relates to the switchable transitions of the condensates defined by the authors to undergo between gel /liquid /dispersed states.

This reviewer feels that the term “dispersed” is inappropriate. It basically presents a fully dissociated state of the condensates constituents. The authors are suggested to exchange the term “dispersed” to “separated constituents” throughout the paper.

Otherwise, this is a nice and interesting paper that is recommended for publication in Nat. Communications.

Our responses

Indeed, UV-irradiate DNA motifs remain around the irradiated region, rather than completely diffuse away to even distribution. We thus agree that the UV-decondensed state does not refer to the ‘dispersed’ state as Reviewer and other readers might imagine. We highly appreciate this comment.

Nevertheless, we still hesitate to use “separated constituents” as a label to that state. First, the term “separated” would be likely to mislead the readers, because ‘phase-separated’ states, *i.e.*, ‘gel’ and ‘liquid’ states, might come to their mind. Second, “gel” and “liquid” refer to the macroscopic states, while “separated constituents” point to the underlying components.

Instead, we would like to use “dissociated” to address the nuances. “Dissociated” is a quite neutral word referring to the non-binding motifs, with no implication of complete dissolution throughout the field.

In this revision, we have:

- | | |
|--|--|
| | - replaced “dispersed” with “dissociated” throughout the sentences in the main manuscript and Supplementary Information.- (p. 32, bottom) replaced “until complete dispersion” with “until disappearance”.- Accordingly, updated Fig. 1, Fig. 2, Fig. 3, Fig. 8, and Extended Data Fig. 3 (in the main) and Supplementary Fig. 8, where ‘dispersed’ states were mentioned. |
|--|--|

Reviewer #3

Thank you for professional comments.

To minimize the potential reader's cumbersome navigation, we further updated the manuscript as below by deleting the inflated statements.

For clarity, the sentences and words newly added in the revision are colored in red, and those deleted in gray.

Comments	Our responses
I inspected the revised version of the manuscript now entitled “Remote-controlled mechanical and directional motions of photoswitchable DNA condensates” by Udonno et al. I find it much improved with respect to the previous version, and I am in favor of publishing it. I must however say that the inflated style of abstract and introduction still annoys me. What is the purpose of the 1-3 requirements for “microfluidic actions”? It is clear that they have no part in the MLO example given right after in the same section.	We appreciate this professional comment. The 1–3 requirements are the design guide, generally acceptable, for creating the remote-controlled micro-scale fluid, mentioned in the beginning of Introduction. These principles run throughout this study. On the other hand, the MLO examples are to give biological strategy on how their molecular-level interactions lead to their functions, organization, and properties, which are available on larger scales. Nevertheless, we can agree that the reader may sweat to bridge these two concepts consistently. In the revised manuscript, we have made clear that 1–3 requirements are our design principle, and that these features can be met by DNA nanotechnology and DNA condensate by modifying: (pp. 3–4, bottom) “A rational design principle is to translate molecular-level control into larger-scale microflow behaviors, which is realized through the following three mechanisms: (1) ...Dynamics programmability based on molecularly encoded information.” Also, we have deleted the next MLO description (p. 4, upper) “One instructive example of the programmed dynamics is observed in liquid condensates arising from the liquid-liquid phase

	separation of biomolecules in living cells.... The fluid property changes are thus linked to many diseases resulting from the dynamic molecular exchange malfunctions.” Further, The MLO statements just deleted have been moved to the description of DNA liquid condensate, and made concise, as (p. 4, bottom) “...as model systems of biological phase-separated liquid condensates, which...and links to diseases”. Lastly, the bottom-up programmability of DNA liquid condensates has been made clear, as (pp. 4–5, bottom) “The molecular-level base-sequence design allows DNA liquid condensates to realize...and physical properties.”
Also, what is described is not a real “photocontrollable locomotion” on the micron scale in the way everybody would desire it to be.	We have deleted the citation related to “photocontrollable locomotion” (p. 3, lower), not to mislead the readers to think of “photocontrollable locomotion” as the readers desire it to be, where droplets are sliding backward and forward on the substrates.
I see no way in which these findings will turn into “programmable, remote-controlled microfluidic systems, including DNA-fueled selective transport, artificial micromuscles, and microstructure folding.” and I dislike science too full of hypes, in which the reader must navigate between claims to get the real core findings. Nevertheless, since the findings are interesting, and since my duty is not to address the publication style (this is to the editors to decide), I give my green light to the publication.	We have deleted the following statements suggesting potential applications that await future work, not to mislead the readers to navigate through specific core claims: (Abstract, p. 2) “These findings will emerge as...and microstructure folding” (Results & Discussion, p. 25, middle) “...capable of e.g.,...artificial muscles” (Conclusions, pp. 26–27) “The collapse mode may...on microfluidic platforms.”